# TRANSLIGHT: IMAGE-GUIDED CUSTOMIZED LIGHTING CONTROL WITH GENERATIVE DECOUPLING

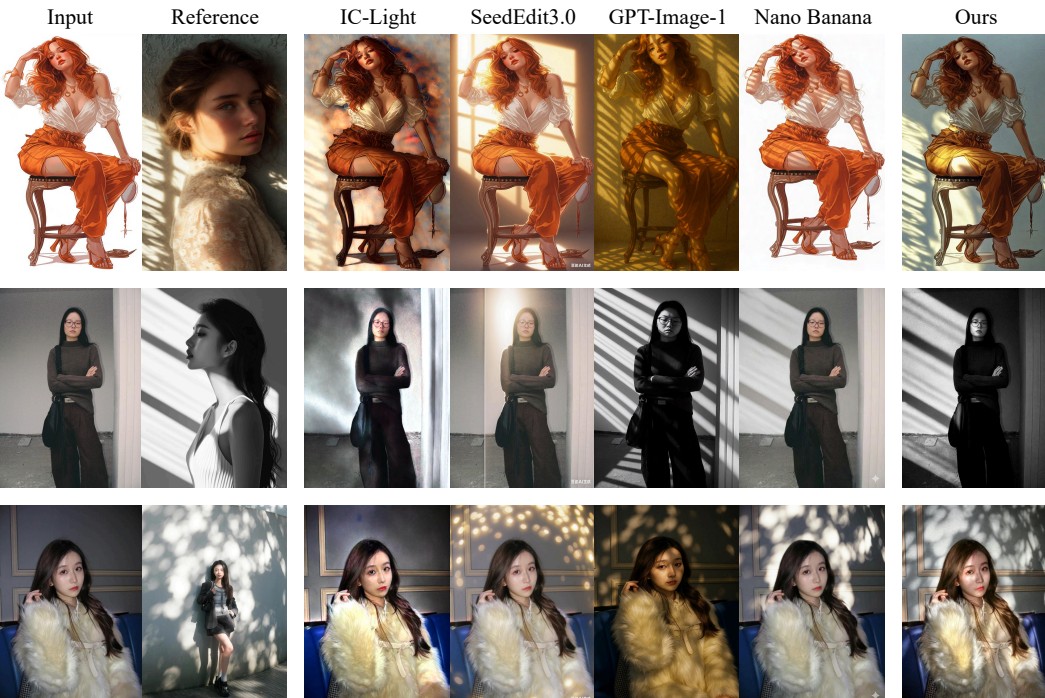

Figure 1: **Transfer light effects from reference to input image.** Our TransLight is a novel method that high-fidelity transfers light effects with obvious geometric structure from reference images to input images. The background conditioned model of IC-Light (Zhang et al., 2025) exhibits an overall darker tone and may introduce content from the reference image into the resulting images. SeedEdit3.0 (Wang et al., 2025) can not perceive the type, direction and structure of light effects in the reference image. GPT-Image-1 (OpenAI, 2025) partially understands the light effects in the reference image, but alters the human ID. The interaction between the character and the light effect in the transferred result of Nano Banana (Google DeepMind, 2025) is not natural enough.

## ABSTRACT

Most existing illumination-editing approaches fail to simultaneously provide customized control of light effects and preserve content integrity. This makes them less effective for practical lighting stylization requirements, especially in the challenging task of transferring complex light effects from a reference image to a user-specified target image in portrait photography scenes. To address this problem, we propose **TransLight**, a novel framework that enables high-fidelity and high-freedom transfer of light effects. Extracting the light effect from the reference image is the most critical and challenging step in our method. The difficulty lies in the complex geometric structure features embedded in light effects that are highly coupled with content in real-world scenarios. To achieve this, we first present Generative Decoupling, where two fine-tuned diffusion models are used to accurately separate image content and light effects, generating a newly

curated, million-scale dataset of image–content–light triplets. Then, we employ IC-Light as the generative model and train our model with our triplets, injecting the reference lighting image as an additional conditioning signal. The resulting TransLight model enables customized and natural transfer of diverse light effects. Notably, by thoroughly disentangling light effects from reference images, our generative decoupling strategy endows TransLight with highly flexible illumination control. Experimental results establish TransLight to successfully transfer light effects with geometric structure across disparate images, especially in portrait photography, delivering more customized illumination control than existing techniques and charting new directions in illumination harmonization and editing.

# 1 INTRODUCTION

Recent rapid iteration of image illumination editing methods (Zhang et al., 2021; Dastjerdi et al., 2023; Choi et al., 2024; Fortier-Chouinard et al., 2024; Kim et al., 2024; Ren et al., 2024; Phongthawee et al., 2024; Zeng et al., 2024a; Jin et al., 2024; Zhang et al., 2025; Magar et al., 2025; Chaturvedi et al., 2025) has led to significant advancements in manipulating lighting attributes within the field of image editing. Relighting of human portraits, as a specific branch in this area, holds broad application prospects and high practical value. Existing human portraits relighting approaches can generally be divided into two categories: visual-guided and text-guided illumination adjustment. Visual-guided methods (Kim et al., 2024; Ren et al., 2024; Chaturvedi et al., 2025) often employ high dynamic range (HDR) maps or additional background images to composite human portraits into new lighting environments. However, this typically leads to a loss of content integrity in the original image. Text-guided methods (Zhang et al., 2025; Cha et al., 2025) accept textual descriptions of the desired light effects as input to perform image relighting. It is evident that textual descriptions alone are insufficient to achieve customized control over lighting properties such as direction, position and geometric structure. Although these methods can achieve striking visual effects, the application scenarios for relighting human portraits through background replacement or text prompts remain limited. In real-world image editing scenarios, users may prefer personalized lighting modifications (e.g., adding or removing specific light effects) while preserving the original content of the image. For example, a user might request: *"Add a rainbow lens flare similar to the reference image to my outdoor travel photo."* Existing relighting approaches often struggle to fulfill such customized lighting stylization requirements.

In this work, we focus on the highly challenging task of transferring light effects from a reference image to a target image in portrait photography scenes. To achieve this, we first need to address two main challenges: (1) *decoupling light effects from content to prevent content leakage* and (2) *constructing a suitable data structure for light effect transfer training.* Although this task is broadly similar to well-established style transfer, both aiming to replicate specific attributes from one reference image onto another image, it requires additional consideration of geometric information such as the direction, structure, and position of the light effects. This leads to a stronger coupling between the light effects and the content, making many existing strategies (Wang et al., 2024a;b; Xing et al., 2024; Jeong et al., 2024) designed to mitigate content leakage in style transfer task less applicable to our case. Aiming at the challenges mentioned above, we propose **Generative Decoupling** as the solution. Specifically, we fine-tune two diffusion models using natural images without light effects and light material images, enabling them to remove and extract light effects from input images. After sufficient training, our light extraction model is capable of effectively isolating the most prominent light effects without incorporating any other content. This largely avoids the issue of content leakage in most cases. Moreover, we construct over one million image-content-light triplets for light effect transfer task training using our generative decoupling strategy.

After that, we propose TransLight, a novel method capable of transferring light effects with obvious geometric structure from a reference image to another image. It is worth noting that we focus on visually perceptible light effects with distinct geometric structures, such as light beam in Tyndall Effect, iridescent halo, and lens flare, among others. We employ IC-Light (Zhang et al., 2025) as the fixed generative network and train our model using the image-content-light triplets. During inference, the target light effect is decoupled from the reference image, effectively mitigating content leakage and enabling greater transfer flexibility. Users can freely adjust the transferred result by modifying the position and orientation of the light effect image. Experimental results demonstrate

UJpe: W1, Q1

that our TransLight possesses a powerful capability for customized light effects transfer. It can retain the background information of the source images while naturally incorporating target light effects from reference images as shown in Figure 1. Moreover, our method achieves a Light-FID score of 6.02, outperforming the SOTA illumination editing method IC-Light, which obtains 10.05.

The main contribution can be summarized as follows:

- We propose **Generative Decoupling** to decouple content and light in natural images using two diffusion models. Among them, light removal model eliminates prominent light effects from natural images, *e.g.*, lens flares, Tyndall effects, *etc*. Light extraction model is the first capable of separating light effects from images without retaining any original content.

- We construct an image-content-light triplet generation pipeline based on our generative decoupling strategy. We first select images with prominent light effects from a massive database, and then use our light removal and light extraction models to generate the corresponding content and light effects images, respectively. After filtering step, we obtain over one million training samples.

- We introduce **TransLight**, a novel image-guided method that allows accurate transfer of light effects with obvious geometric structure from a reference image to a target image in portrait photography scenes. Our method supports translation, flipping, and numerical scaling of the extracted light effects, enabling adjustments to the position, orientation, and intensity of the light effects in transferred result. Experimental results show that our TransLight achieves visually striking light effects across a wide range of scenarios.

## 2 RELATED WORK

### 2.1 IMAGE STYLE TRANSFER

Image style transfer aims to transfer the style of a reference image to the target content image (Chung et al., 2024; Gao et al., 2024; Wang et al., 2023b; Zhang et al., 2023b; Frenkel et al., 2024; Wang et al., 2024b; Qi et al., 2024; Lei et al., 2025; Xing et al., 2024). This process is fundamentally similar to light effects transfer, which aims to replicate the intrinsic attributes of a reference image onto another image, while preserving the content consistency of the target image. In this task, a well-studied problem is how to disentangle the content and style of the reference image to avoid content leakage. StyelAligned (Hertz et al., 2024) proposes replacing full attention with shared attention during style injection to reduce content leakage. FreeTuner (Xu et al., 2024) employs a disentanglement strategy that separates the generation process into two stages to effectively mitigate concept entanglement. InstantStyle (Wang et al., 2024a) and Swapping self-attention (Jeong et al., 2024) minimize the introduction of content information by injecting style features into specific attention layers of the generation network. The aforementioned methods are specifically designed to disentangle abstract stylistic attributes from images and have proven effective in mitigating content leakage. However, they lack the capability to precisely perceive the geometric and structural characteristics of light effects, making it difficult to extract light. As a result, these approaches are not well-suited for the task of light effects transfer.

### 2.2 LIGHT REMOVAL

Light removal aims to eliminate unwanted illumination from images while meticulously preserving the underlying scene content. The most representative application is the removal of flares from images (Dai et al., 2023; Wang et al., 2023a; Jin et al., 2022; Qiao et al., 2021). (Wu et al., 2021) collected 2K real reflective flare data using a camera to train the first general lens flare removal network. After that, (Dai et al., 2022) further constructed 7K synthetic flare data to reduce the domain gap with real-world scenarios. (Qiao et al., 2021) propose the first unpaired flare removal dataset and present a deep framework with light source aware guidance for single-image flare removal. Recently, Illuminet (Xu et al., 2025) proposed a two-stage pixel-to-pixel generative model based on the U-Net (Ronneberger et al., 2015a) model, that achieved both image flare removal and image light enhancement. In addition, LuminaBrush (Team, 2024) convert images to "uniformly-lit" appearances by fine-tuning the DiT (Peebles & Xie, 2023) model through LoRA (Hu et al., 2022). While these methods work well in specific situations, their limited training dataset makes it hard for them

to perform satisfactorily in complex scenes, which means they can not adequately support the our decoupling requirements.

### 2.3 LIGHTING CONTROL WITH DIFFUSION MODELS

Diffusion models have recently found widespread application in highly challenging tasks that demand precise control over image illumination, including portrait relighting and indoor lighting adjustment. For portrait relighting, methods such as DiFaReli (Ponglertnapakorn et al., 2023), Lite2Relight (Rao et al., 2024), and Holo-Relighting (Mei et al., 2024) specifically target the relighting of human faces. Relightful Harmonization (Ren et al., 2024) and Total Relighting (Pandey et al., 2021) manipulate the illumination of the foreground portrait using background conditions. SwitchLight (Kim et al., 2024) controls portrait illumination by using high dynamic range (HDR) images as conditions. Text2Relight (Cha et al., 2025) generates relighted portrait using text prompt while keeping the original contents. Furthermore, IC-Light (Zhang et al., 2025), trained extensively on large-scale datasets using the principle of light transport consistency, achieves impressive capabilities in text prompt and background controllable portrait illumination generation. For indoor lighting adjustment, common operations include introducing new light sources or regulating the illumination intensity of those already present in a scene (Zhang et al., 2024; Wang et al., 2022; Magar et al., 2025). SpotLight (Fortier-Chouinard et al., 2024) control the local lighting of an object through a coarse shadow. By training a ControlNet (Zhang et al., 2023a), ScribbleLight (Choi et al., 2024) controls image light effects based on user scribbles. Latent Intrinsics (Zhang et al., 2024) and LumiNet (Xing et al., 2025) are capable of modifying the illumination states of input images based on the light effects from a reference image. DiffusionRenderer (Liang et al., 2025) use specified environment map to accurately estimates geometry and material buffers and generates photorealistic images.In addition, in the field of video generation, Light-A-Video (Zhou et al., 2025) and Re-lightVid (Fang et al., 2025) use IC-Light to add artistic lighting and shadow effects to the video generation process, and VidCRAFT3 (Zheng et al., 2025) control the light direction using per-frame 3D vector in Cartesian coordinates. While the aforementioned light control methods achieve good results in specific scenarios, their customized control over light effects is highly limited. They cannot precisely and comprehensively adjust details such as light position, and shape while preserving image content. Our TransLight aims to provide a solution.

UJpe: W1, W2

## 3 METHOD

In this section, we first provide a clear task definition for the light effect transfer task: *Given a user input image $I$, a reference image $R$ and user control parameters $p$ (e.g., translation, flip, intensity), finally obtain the edited result image $I_L$.* The most ideal generation result is that $I_L$ has the same light effects as the reference image $R$, mainly manifested in the consistency of the geometric structure and the type of the light effect in the two images. Moreover, $I_L$ should maintain complete content consistency with the input image $I$. The light effect transfer task we define is formally similar to the illumination condition transfer task in LumiNet (Xing et al., 2025), yet exhibits clear distinctions in many aspects. We provide a detailed discussion in the supplementary material.We systematically illustrate the overall framework for achieving the task of light effect transfer in Figure 2. This framework comprises three key modules: the model training of generative decoupling, the image-content-light triplet construction pipeline, and the training of our TransLight.

UJpe: W1, W2

### 3.1 GENERATIVE DECOUPLING

We use two diffusion models to decouple the image content and light effects, which we refer to as generative decoupling. We design a simple yet effective strategy to train our light removal and light extraction models. Figure 2 (a) shows our training process.

**Source data collection.** We first employ InternVL2.5 (Chen et al., 2024) to select 1M images without light effects from a massive database. Then, we collect 100K light material images, all featuring pure black backgrounds. Specifically, we generate approximately 90K light material images using FLUX.1-schnell (Labs, 2024) and collect an additional 10K real images from public datasets (Dai et al., 2022; Wu et al., 2021). The details of the prompts used in the image collection process are given in the appendix.

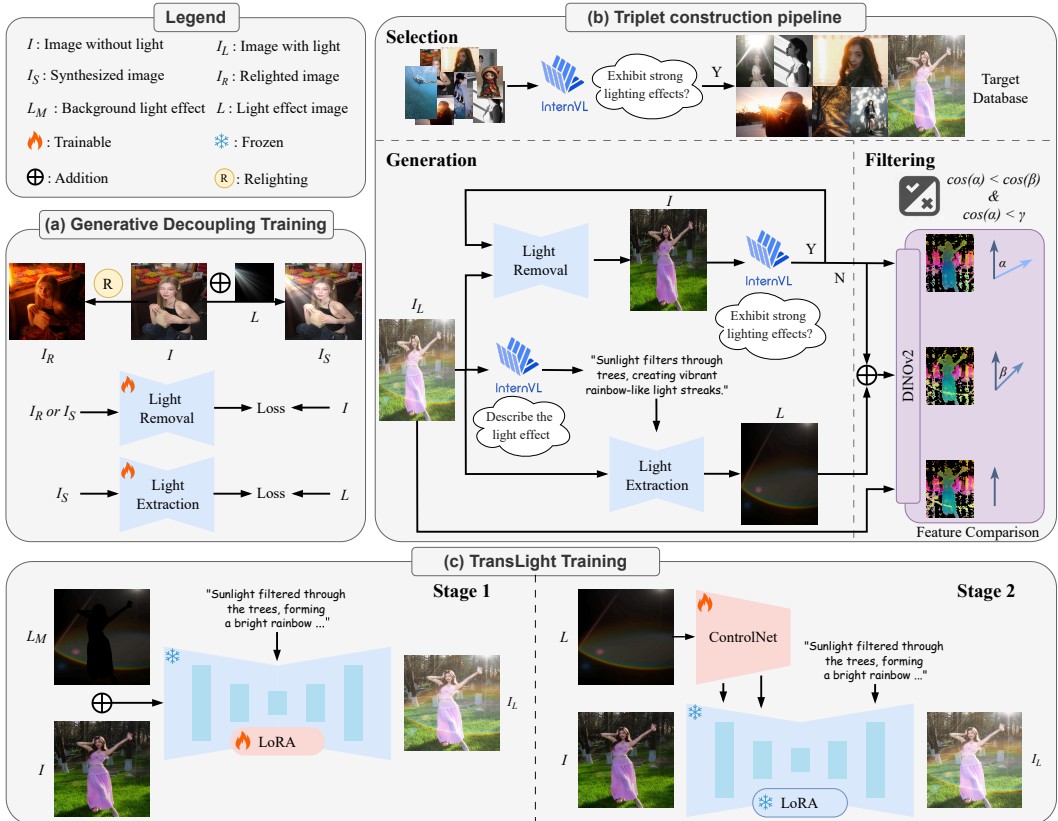

Figure 2: **Overall framwork.** (a) We fine-tune two diffusion models initialized with the weights of IC-Light. During training, the input synthesized image $I_S$ is the direct addition result of no light image $I$ and light material image $L$. (b) This pipeline involves selecting relevant data using a vision language model, decoupling image content and lighting via light removal and extraction models, and filtering out poor generation results. (c) Our TransLight training consists of two stages: LoRA fine-tuning for content-preserving lighting editing and ControlNet training to inject light effects.

**Training input condition image construction.** During training, the input condition images provided to the model are those containing prominent light effects. However, directly using real-world images would make it difficult to obtain effective supervision signals. Therefore, we simulate such lighting-affected images by applying preprocessing to the images from the source data. We apply two distinct preprocessing methods to obtain $I_S$ and $I_R$. The synthesis of no light image $I$ and light material $L$ into $I_S$ involves no complex operations and can be expressed by $I_S = aI + bL$, where $a, b \sim U(0, 1)$. This simple and direct operation is designed to simulate real-world scenarios in which lighting effects exhibit prominent geometric structural characteristics. In addition, to simulate strong lighting contrasts such as pronounced shadows and highlights commonly observed in real-world scenes, we relight the image $I$ to generate the $I_R$. We use relighting model IC-Light and image editing methods FLUX.1-Kontext-dev (Labs et al., 2025) and Qwen-Image-Edit (Wu et al., 2025) as the relighting methods, and use diverse prompts to obtain $I_R$.

**Training strategy.** We fine-tune two diffusion models to obtain the light removal and light extraction models. We initialize the U-net (Ronneberger et al., 2015b) architecture using the weights from IC-Light (Zhang et al., 2025). The training input of the U-net (Ronneberger et al., 2015b) architecture is the 8-channels concatenation representation of noisy latent and input condition image latent. For light removal model training, the input condition image consists of two parts: synthesized image $I_S$ and relighted image $I_R$. The prediction target is image $I$. For light extraction model training, the input image is $I_S$ and the prediction target is light effect image $L$. Experimental results demonstrate that even with simple synthesis operations, our fully trained light extraction model achieves good performance in real-world scenarios.

## 3.2 IMAGE-CONTENT-LIGHT TRIPLET CONSTRUCTION PIPELINE

To make light effects transfer possible, we construct a large volume of image-content-light triplets. As shown in Figure 2 (b), our triplet construction pipeline contains three steps.

**Selection.** Initially, we select target data from a massive proprietary database. We employ InternVL2.5 for this large-scale batch selection. We use highly stringent prompts to instruct the vision language model to determine if images contain strong light effects.

**Generation.** We use light removal model and light extraction model to decompose an image with strong light effects into an image containing only content and an image containing only light effects. For the light removal process, we use InternVL2.5 to evaluate the generated results. If the results do not meet requirements, we perform a second round of light removal. In the light extraction process, we also use InternVL2.5 to generate a description of the image's light effects. This description, when combined with the image as input to the light extraction model, notably enhances the model's ability to separate light.

**Filtering.** We perform similarity-based filtering on the generated results to improve training data quality, discarding samples that exhibit notably poor generation outcomes. Specifically, after decoupling the input image $I_L$ into a no light image $I$ and a light effect image $L$, we obtain $I_S$ using the simple synthesis strategy. Then, we extract features from $I_S$, $I_L$, and $I$ separately using the DINOv2 (Oquab et al., 2023), and compute the cosine similarity of these features. The specific calculation method is as follows:

$$cos(\alpha) = Sim_{cos}(\mathcal{D}(I_L), \mathcal{D}(I)), \tag{1}$$

$$cos(\beta) = Sim_{cos}(\mathcal{D}(I_L), \mathcal{D}(I_S)), \tag{2}$$

where $\mathcal{D}$ stands for DINOv2 and $Sim_{cos}$ stands for cosine similarity. During our triplet construction, we aim for a significant degree of light effect removal, which necessitates that the similarity between $I$ and $I_L$ falls below a threshold $\gamma$. Concurrently, we desire a salient and clean separated light effect, thus expecting an increased similarity between $I_S$ and $I_L$. This means that the feature similarity among these images should satisfy the following expression:

$$(cos(\beta) > cos(\alpha)) \land (cos(\alpha) < \gamma), \tag{3}$$

where $\gamma$ is set to 0.98. We set $\gamma = 0.98$ for the trade off between data quality and data quantity. The larger the threshold $\gamma$ is, the more data remains after filtering, but the data quality varies. Conversely, the smaller the threshold $\gamma$ is, the less data remains after filtering, but the data quality is higher. 

`67f2: W2`

## 3.3 TRANSLIGHT

Figure 2 (c) shows the training process of our TransLight. We adopt a two-stage training strategy because the training objectives of the two modules differ. The lora fine-tuning in the first stage is to enable the model to keep the background unchanged. The second stage of training is aimed at providing complete light effect images and enhancing the degree of conditional injection. 

`67f2: W1;`
`Y35F: Q1`

**Training stage 1.** First, we use the SD1.5 (Rombach et al., 2022) version of the IC-Light (Zhang et al., 2025) as our generation network. This is done to fully leverage its powerful illumination editing capabilities, which were acquired through training on large-scale datasets. We freeze the parameters of the generation network during training. We first perform LoRA (Hu et al., 2022) fine-tuning on the generation network, with the goal of enabling the model to preserve the background unchanged during the generation process and enhance its content consistency. The input during this stage is a simple addition of the content image and the background light effects image. We concatenate the latent-space features of this single image with the latent noise along the channel dimension and feed the result into the first layer of the U-Net, as in IC-Light. This stage aims to enable the model to generate realistic images from composite inputs based on content and light effects image. Additionally, we use only the background portion of the light effect to avoid affecting the foreground person. Specifically, we use background removal model BiRefNet (Zheng et al., 2024) to obtain the foreground mask $M$ for the content image. The background light effects image $L_M$ is equivalent to $L \times (1 - M)$. 

`SUh4: Q1`

**Training stage 2.** In the stage 2, we inherit and fix the LoRA weights trained in the stage 1, and train an additional ControlNet (Zhang et al., 2023a) to inject the light effects image into the generation

Figure 3: **Visualizations of our generative decoupling.** Our light removal model can eliminate light effects from images while preserving the underlying content unchanged as shown in the second line. Additionally, our light extraction model is capable of isolating the light effects from the image without introducing other extraneous objects.

process. To ensure the effectiveness of the ControlNet training, we only use the content image with the light effect removed as the additional channel input for the U-net structure during this stage. The input image for ControlNet is the extracted light effect image.

**Inference stage.** The complete inference pipeline requires three components: the light extraction model, LoRA weights, and ControlNet weights. The light extraction model first extracts light effects from a reference image, which are then fed into the ControlNet. After that, the background light effect is directly added to the content image, providing a more direct and effective condition control. After encoding into latent space via the VAE encoder (Kingma et al., 2013), the result is concatenated channel-wise with Gaussian noise and fed into the generator, yielding the final output through multi-step denoising. Notably, our TransLight allows for flexible adjustment of the following light effect attributes:

- **Position.** Apply horizontal or vertical translation to the light effect image.
- **Direction.** Perform flipping or rotating operations on the light effect image.
- **Intensity.** Modify the multiplicative coefficient of the light effect image overlaid on the content image.

## 4 EXPERIMENTS

### 4.1 TRAINING DETAILS

**Generative Decoupling.** We fine-tune the SD1.5 (Rombach et al., 2022) version of IC-Light (Zhang et al., 2025) model. For both light removal and light extraction model training, we use the AdamW optimizer (Loshchilov & Hutter, 2017) with the learning rate of 1e-5. We train two models on 8 $\times$ L40s GPUs with a batch size of 128. We only use MSE loss to train our models as the normal SD1.5 training. For light removal model training, the training input image consists of two parts: the synthesized image $I_S$ and the relighting image $I_R$. During training, the sampling ratio of these two parts of image is $1:1$, and the ratio of relighting images generated by IC-Light, FLUX.1-Kontext-dev and Qwen-Image-Edit is $2:1:1$. We use a fixed prompt and train the light removal model for 2K iterations. For light extraction model training, the input image is the synthesized image $I_S$ and the optimization target is to generate the light effects contained in the normal image. To enhance the robustness of the light extraction model, when synthesizing $I_S$, we perform appropriate data augmentation operations on L, such as flipping, cropping, and random masking, etc. We use the caption corresponding to light effect image as prompts and train the light extraction model for 26K iterations. The caption is obtained using InternVL2.5.

Table 1: **Quantitative results.** IC-Light uses its background conditioned model, with ground truth serving as the background input.

| Method | PSNR ↑ | SSIM ↑ | LPIPS ↓ |
|--------|--------|--------|---------|
| IC-Light | 15.34 | 0.6784 | 0.2756 |
| Ours | 19.58 | 0.7931 | 0.1982 |

**TransLight.** We update the parameters of LoRA and ControlNet using the AdamW optimizer with a learning rate of 1e-5 for the two stages. Considering that our training data scale is on the order of millions, we set the rank of LoRA to 128 and train it for 10K iterations in stage 1. The training process for stage 2 requires 40K iterations. We train our TransLight on $16 \times$ L40s GPUs with a batch size of 128.

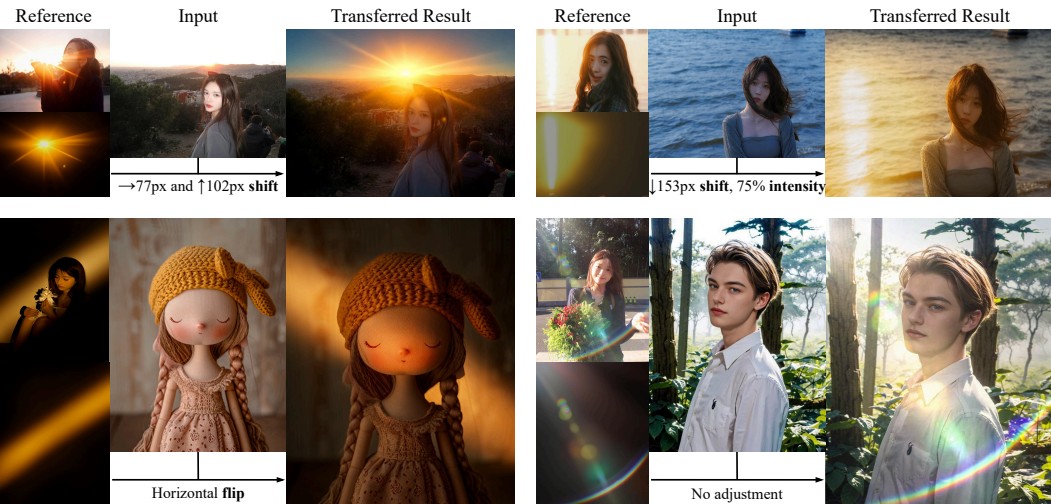

Figure 4: **Visualizations of the specific details about the light transfer process.** Our TransLight uses the generative decoupling to break down such a difficult task. We first extract the light effect using our thoroughly fine-tuned light extraction model, as shown in the reference image below, and then composite it onto the user-specified image. Our TransLight allows flexible control over the position and direction of the light effects, thereby yielding visually authentic results with a high degree of freedom.

### 4.2 PERFORMANCE OF GENERATIVE DECOUPLING

During the process of constructing triplets using generative decoupling, all the source images with obvious light effects are selected from the massive database using InternVL2.5. When quantitatively evaluating the light effect removal capability of the light removal model, we also use InternVL2.5 to judge the generated images, and use this as the basis for determining whether the removal is qualified. We evaluate the light effect removal performance on 500 real-world images, achieving a success rate of 85.8%. To further intuitively demonstrate the performance of our generative decoupling, we present visual results of decoupled generations in Figure 3. Our method achieves high-quality decoupled generation on images with various types of light effects across different scenes.

### 4.3 PERFORMANCE OF TRANSLIGHT

In this section, we present the generated results of our method and comparisons with other approaches using both subjective and objective evaluation methods. For objective comparisons, we first use metrics such as Peak Signal-to-Noise Ratio (PSNR), Structural Similarity Index (SSIM), and Learned Perceptual Image Patch Similarity (LPIPS) to evaluate the similarity between a single generated image and ground truth. Considering that the ground truth of the transfer result can not be obtained during real scene reasoning, we use the real image itself with as both the ground truth and the reference image when calculating the quantitative indicators. Therefore, using the background replacement relighting model in the visual-guided methods can, to a certain extent, achieve the transfer of light effects with obvious geometric structures to the content image while retaining the background content of the original image. It should be noted that, since representative background

Table 2: **Light-FID.** "Content" in the first row refers to result of lighting-removed content image. "Content+Light" refers to the result formed by directly adding content and light effect maps. IC-Light uses its background conditioned model, with ground truth serving as the background input.

| Method | Light-FID |
|---|---|
| Content | 10.61 |
| Content+Light | 8.40 |
| IC-Light | 10.05 |
| Relighted data | 7.72 |
| Unfiltered data | 6.49 |
| Only LoRA | 7.94 |
| Only ControlNet | 8.44 |
| LoRA+ControlNet (joint) | 7.42 |
| Ours | 6.02 |

Table 3: **User study.** For "IC-Light (bg)", we use the reference image as the background input to the background-conditioned IC-Light. For "IC-Light (prompt)", we use the description derived from the reference image. For Nano Banana and Seedream 4.0, we provide both the input image and the reference image to the image editing model simultaneously and use appropriate prompt to complete the light effect transfer task.

| Method | Consistency Pass Rate | |
|---|---|---|
| | Light | Content |
| IC-Light (prompt) | 4.0% | 12.0% |
| IC-Light (bg) | 8.4% | 8.8% |
| Nano Banana | 39.5% | 61.3% |
| Seedream 4.0 | 32.4% | 31.1% |
| Ours | **84%** | **73.8%** |

replacement relighting methods such as Total Relighting (Pandey et al., 2021) and Relightful Harmonization (Ren et al., 2024) are not publicly available, we adopt the background conditioned model of IC-Light as the comparison baseline. We select 1000 triplet samples from the large-scale dataset constructed using the generative decoupling strategy, which are not involved in the training process. Our TransLight uses the light effect images from triplets for condition generation. Regarding IC-Light, we apply its background conditioned model, using the ground truth as the background input. As shown in Table 1, our method clearly outperforms IC-Light.

In addition to the above-mentioned similarity metrics computed on individual images, we believe that calculating the distributional distance between the generated images and the real images provides a more objective evaluation of the overall model performance. Therefore, we sample 12,000 triplets that are not used during training. We compute the Fréchet Inception Distance (FID) (Heusel et al., 2017) to evaluate the light transfer capability of the model, which we refer to as **Light-FID**. The Light-FID reflects the distance of the data distribution between the transfer result images and the real images with obvious light effects. We report the result in Table 2. When performing inference using the background conditioned model of IC-Light, the ground truth is also directly used as the background input. Our method demonstrates significant advantages over IC-Light. Moreover, using only the light effects from the light extraction model and adding them to the content image yields better results than IC-Light.

For subjective comparison in real-world application scenarios, we show some visualizations in Figure 1. To fully highlight the advantages of our TransLight, we also show the result of the SOTA relighting method IC-Light, alongside the closed-source SOTA image editing methods SeedEdit3.0 (Wang et al., 2025), GPT-Image-1 (OpenAI, 2025) and Nano Banana (Google DeepMind, 2025).The reference image is used as background input for IC-Light's background conditioned model. For SeedEdit3.0, GPT-Image-1 and Nano Banana, we directly provide both input image and reference image to the model simultaneously. We experiment with different prompts to instruct the model and select the best output from the results. As illustrated in Figure 1, current relighting and image editing methods are unable to fulfill our expected light effect transfer requirements. IC-Light incorporates the content of the reference image into the generated results, yet the overall style tends to be darker. Although SeedEdit3.0 is capable of modifying the light effects in the target image, it fails to transfer light effects according to user preferences from a reference image. The generated results of the GPT-Image-1 show noticeable changes in the character appearance. The light effect transfer capability of Nano Banana most closely resembles that of our proposed approach, as it can understand the type and structure of light effects from the reference image. However, the visual results indicate weak interaction between the light effects and the human subjects, resulting in a perceived lack of realism. Moreover, free adjustment of light effects position and direction enables more natural integration with the target image, resulting in visually superior

outcomes as shown in Figure 4. Our TransLight successfully implements image-guided customized lighting control via transferring light effects from a reference image to a target image.

### 4.4 USER STUDY

We conduct a user study to evaluate the performance of our model on the light effect transfer task. In total, we collect voting responses from 15 participants. We asked the participants to vote respectively from the dimension of light effect consistency and content consistency, and select the results that meet the requirements in each samples. Good light effect consistency means that the light effect in the generated image closely matches that in the reference image. Good content consistency indicates that neither the subject nor the background in the generated image undergoes noticeable changes. For each sample, we present the generation results of five methods: prompt-conditioned IC-Light (Zhang et al., 2025), background-conditioned IC-Light, Nano Banana (Google DeepMind, 2025), Seedream 4.0 (Seedream et al., 2025), and our TransLight, displayed in random order. We show the result in Table. 3. Our TransLight significantly outperforms all competing methods in both light effect consistency and content consistency.

### 4.5 ABLATION STUDY

**Training strategy.** In this part, we conduct ablation studies on the two-stage training strategy. Specifically, we compute the Light-FID scores under three settings: (1) only LoRA fine-tuning, (2) only ControlNet training, and (3) joint two-stage training. The results are reported in Table. 2. Experimental results demonstrate that both stages contribute to improving TransLight's performance. Moreover, due to the discrepancy in their respective training objectives, training the LoRA and ControlNet modules separately yields better results than joint optimization.

**Training data.** We conduct an experiment in which TransLight is trained on relighted data instead. Specifically, we employ the text-conditioned model of IC-Light to relight real images that originally contain no light effects, and then extract the corresponding light effects images from the generated results to form the triplet data. Using this approach, we also generate one million training samples. The Light-FID is reported in Table 2. This result further justify the rationality of our motivation to construct triplets for training by leveraging real-world data through generative decoupling.

Table 4: **Ablation of filtering threshold $\gamma$.** We report the number of training samples and the Light-FID scores corresponding to different values of the threshold $\gamma$.

| $\gamma$ | Training samples | Light-FID |
|------|------|------|
| 1 | ~1.5M | 6.41 |
| 0.99 | ~1.3M | 6.22 |
| 0.98 | ~1.2M | **6.02** |
| 0.97 | ~0.9M | 6.54 |
| 0.96 | ~0.6M | 6.83 |
| 0.95 | ~0.5M | 7.00 |

**Data filtering.** To validate the effectiveness of the third step, filtering in the receipt construction pipeline, we conduct experimental verification. Specifically, we train our model using one million unfiltered triplet samples and compute the Light-FID score as shown in Table 2. Training with filtered data enables our model to perform better. Moreover, the threshold $\gamma$ significantly affects both the quantity and quality of the data. We perform filtering under different values of $\gamma$ and train the model using the resulting eligible images, and the experimental results are presented in Table. 4. As shown in the table, we set $\gamma = 0.98$ for the trade off between data quality and data quantity.

## 5 CONCLUSION

In this paper, we propose TransLight, a novel lighting-control generation method that enables the transfer of light effects from reference images to target images in portrait photography scenes. To achieve this, we introduce generative decoupling strategy, which decouples the content and light effects in real-world images by fine-tuning two diffusion models. Based on this strategy, we construct over one million high-quality triplet samples, which are then used to train our TransLight. Experimental results demonstrate that our method achieves impressive light effect transfer capabilities, enabling flexible customization of lighting-controlled image generation. Our TransLight provides a solid and flexible foundation for future developments in image-guided light effect transfer, and we are highly optimistic about its promising application potential in practical lighting editing tasks.

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

# APPENDIX

## A  DATASET DETAILS

### A.1  LIGHT MATERIAL

During the training of both the light removal and light extraction models, we use approximately 100K light material images, of which 10K are sourced from public datasets (Dai et al., 2022; Wu et al., 2021), and the remaining 90K are generated using the FLUX.1-schnell (Labs, 2024). We employ the following prompt for generation: *Generate a light material image: <random-example>, the background color is pure black, true and natural, do not show anything other than light effects.* The "random-example" refers to a randomly selected prompt from our designed set of diverse descriptions of light effects. We provide some examples below:

- Mystical light beams traversing a clear yet particulate-rich environment, showcasing the diffusion and beauty of light.

- Organic patterns of random color and random shape light forming soft, diffused spots with varied shapes and sizes, as if passing through a complex, uneven medium.

- A single, intense light source creating a bright spot. The light should have a soft glow around it, with subtle gradients transitioning from bright white to deep black. Include lens flares and light streaks for added realism.

- Abstract a small light with curved rainbow effect, no objects, vibrant light streaks, dynamic glow, detailed texture, photorealistic.

- An abstract underwater scene featuring dynamic light rays piercing through a rippling water surface. The surface is textured with subtle waves and ripples. The overall atmosphere is ethereal, focusing solely on the interplay of light and water.

- Abstract warm oval patches of light beams casting soft shadows on a dark background wall, mimicking the dramatic lighting often used in portrait photography. Focus on the interplay of light and shadow without any objects or figures.
- The glow reflected on the lake surface, with its floating light leaping like gold.
- The golden side light projects regular light and dark patterns on the deep black background, suggesting the warmth and depth of the hidden light source.

We provide some examples of light material generated by Flux in Figure 5.

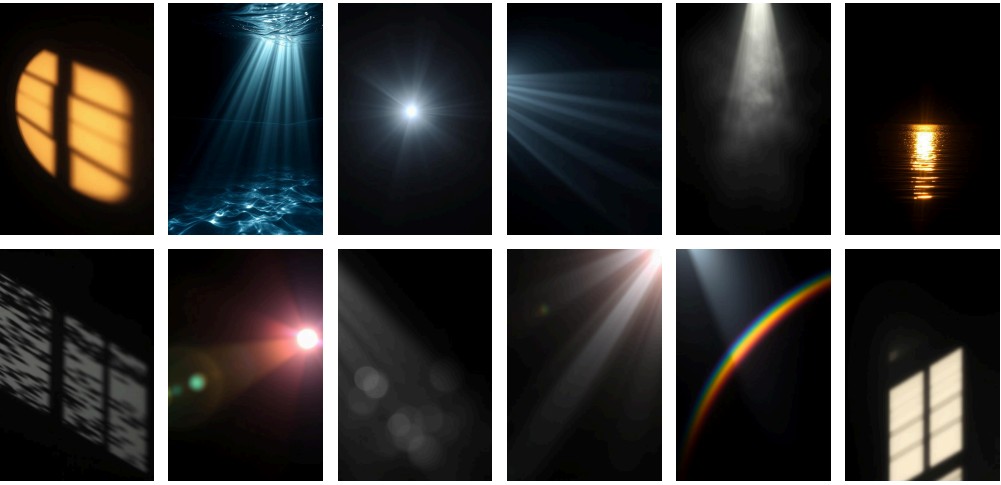

Figure 5: **Visualizations of the light material generated by Flux.**

### A.2  IMAGE-CONTENT-LIGHT TRIPLETS

We first employ the vision-language model InternVL2.5 (Chen et al., 2024) to filter approximately 2 million images with prominent light effects from a collection of over 20 million natural images. Then, we apply our generative decoupling strategy to these datasets for decoupled generation. After the third filtering operation, approximately 1.2 million high-quality triplets that meet expectations are obtained. To accommodate the training requirements of the diffusion model, we use InternVL2.5 to generate brief captions for the images. The triplet data construction process took approximately 12 days, and during this period, we used 16 L40s GPUs for all steps. It takes approximately one day to select images with strong light effects from massive data using VLM. It took 5 days and 6 days respectively to extract the light effects from these images and remove the light effects.

We categorize all triplets in our dataset into nine light-effect types: 1.Light Beam in Tyndall Effect; 2.Dappled Light; 3.Backlighting Effect; 4.Iridescent Halo; 5.Lens Flare; 6.Bright Light Source; 7.Gleaming Water Reflection; 8.Projected Light Patch; 9.Volumetric Light Rays. Using InternVL2.5-8B, we count the number of images associated with each category. The distribution of sample counts across each category is shown in Figure. 6. To provide a more intuitive understanding of the light effects associated with each category, we present representative examples in Figure 7. Unlike traditional indoor scene data, our data mainly consists of portrait photography and natural landscapes, among which portrait data accounts for about 80%

SUh4: W2

SUh4: W2;
UJpe: Q1

### A.3  THE USAGE OF VISION LANGUAGE MODEL

As shown in Figure 2, in the overall framework for implementing the light effect transfer task, the vision language model InternVL2.5 (Chen et al., 2024) plays a very important role. We leverage the InternVL2.5 model to pre-filter a large-scale database, ensuring high-quality inputs for subsequent generative decoupling training and triplets construction.

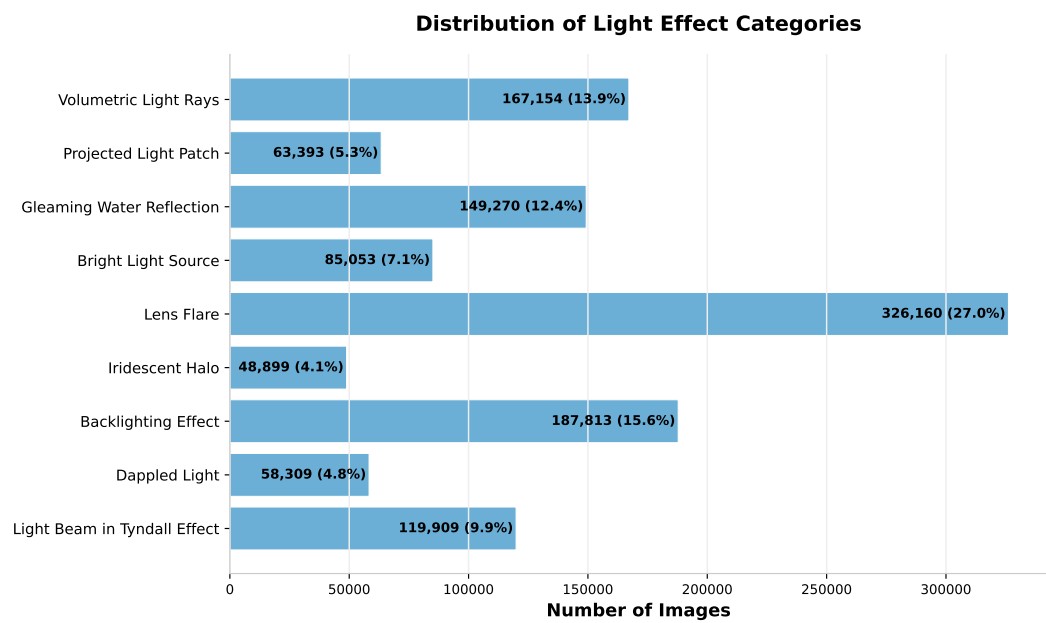

Figure 6: **Light effect distribution.** We categorize light effects into nine classes and perform a statistical analysis on our million-scale triplet dataset.

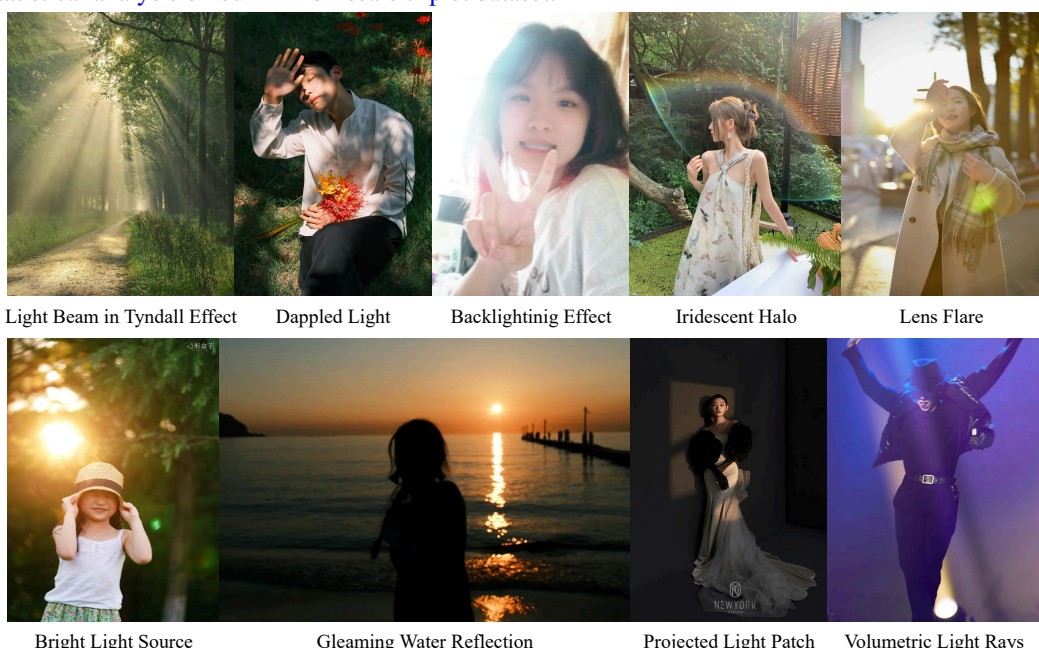

Figure 7: **Examples of different types of light effects.**

We employ the prompt: *"Is there no obvious lighting effect in the image and is the person under uniform lighting?"* to determine whether an image does not contain light effects. We first use this prompt to obtain approximately 1 million images. These images, which exhibit highly similar data distributions in terms of lighting attributes, are used for generative decoupling training process. And this prompt is also applied to judge the validity of the light removal results.

Furthermore, we employ the prompt: *"Does this image exhibit extremely strong lighting effects with dramatic light/shadow contrast, visible light beams/rays, prominent lens flares/light spots, or intense dominant light sources?"* to determine whether an image contains obvious light effects. This prompt is employed to select the source images used for triplet dataset construction.

Table 5: **Quantitative results on the MIIW dataset.** Although our method is not specifically designed for indoor scenes, it remains highly competitive.

| Methods | Labels | RMSE↓ | SSIM↑ |
|---------|--------|-------|-------|
| Input Img | - | 0.384 | 0.438 |
| SA-AE (Hu et al., 2020) | Light | 0.288 | 0.484 |
| SA-AE (Hu et al., 2020) | - | 0.443 | 0.300 |
| S3Net (Yang et al., 2021) | Depth | 0.512 | 0.331 |
| S3Net (Yang et al., 2021) | - | 0.499 | 0.336 |
| Latent-Intrinsic (Zhang et al., 2024)($\sigma$=4) | - | 0.326 | 0.232 |
| Latent-Intrinsic (Zhang et al., 2024) | - | 0.297 | 0.473 |
| RGB-X (Zeng et al., 2024b) | - | 0.256 | 0.476 |
| LumiNet (Xing et al., 2025) | - | **0.180** | **0.647** |
| Ours | - | 0.212 | 0.627 |

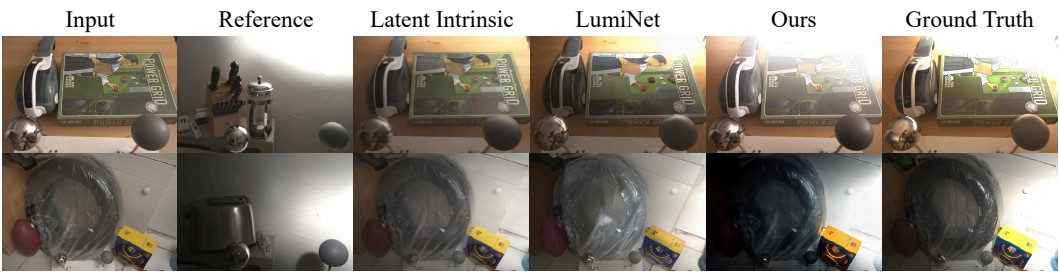

| Input | Reference | Latent Intrinsic | LumiNet | Ours | Ground Truth |

Figure 8: **Visual Comparison on MIIW (Murmann et al., 2019) dataset.**

# B  COMPARISON WITH INDOOR SCENE RELIGHTING METHODS

In the Related Work section of the main text, we briefly review several physically-based relighting methods (Zhang et al., 2024; Xing et al., 2025; Liang et al., 2025) for indoor scenes. Although the light effect transfer task we define bears some formal similarities to the tasks addressed by these methods, our TransLight differs significantly from them in terms of design motivation, application scenario, and prediction targets.

- **Design motivation:** Our TransLight aims to enable the flexible transfer of geometrically structured light effects that are difficult to describe with simple text to other images, whereas physically-based indoor relighting methods primarily seek to achieve improved illumination consistency or coherence within a scene.

- **Application scenario:** Our TransLight focuses on portrait photography scenarios with broad practical applicability, aiming to enhance the aesthetic quality of images, whereas relighting methods such as LumiNet are primarily targeted at indoor scenes, such as bedrooms and living rooms.

- **Prediction targets:** Our TransLight can add customized light effects with distinct geometric structures to the input image. Moreover, our TransLight can also adjust the illumination state of the image by extracting light effects from the reference image to make the transfer result more realistic. For example, it can modify the shadow of the person's face according to the direction and position of the light effect, as shown in the example in the second sample of Figure. 1. The target of previous physically-based methods is to render the input image to a new illumination status, such as converting an "off" desk lamp to an "on" state based on the state of the reference image, or modifying the direction of the shadow from left to right according to a specific Env.Map.

We follow the evaluation method of Latent Intrinsics (Zhang et al., 2024) and conduct zero-shot evaluation of our TransLight on the MIIW dataset (Murmann et al., 2019) to enable a thorough comparison. We randomly select an image and its 12 reference image for light effect transfer

Input    Reference    LumiNet    Ours

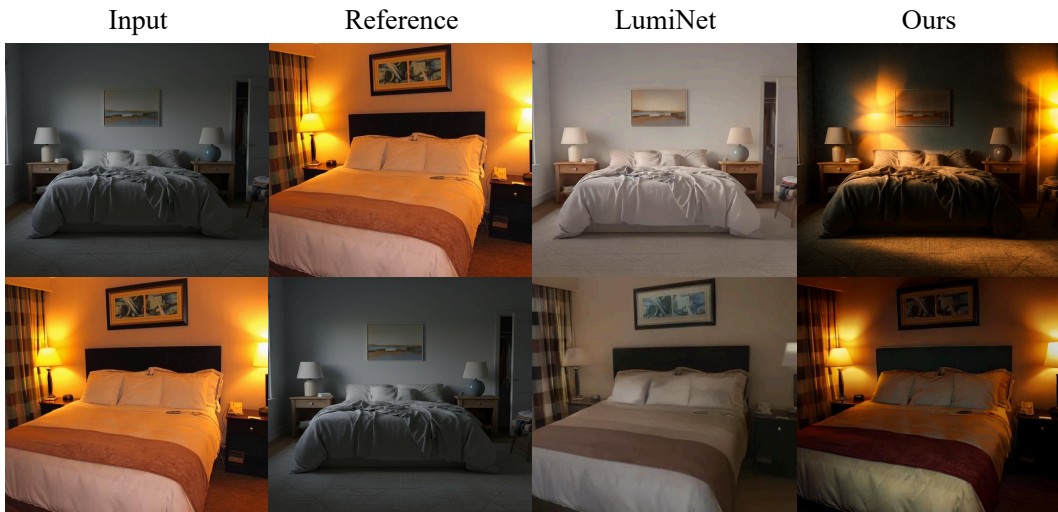

(a) State transfer of lamps in indoor scenes

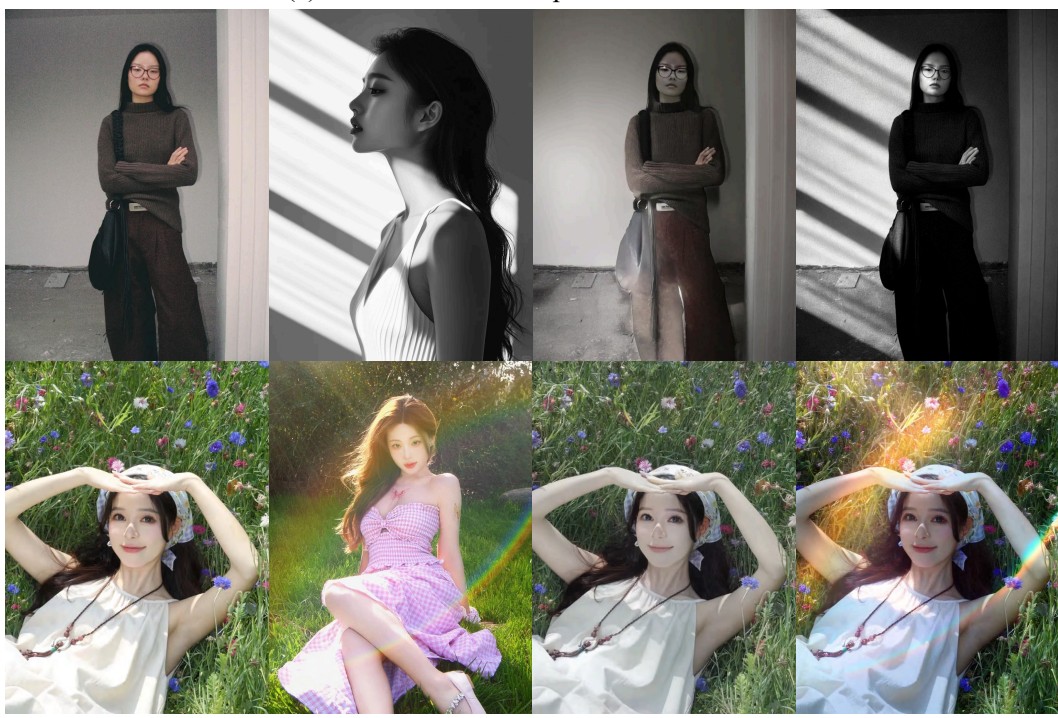

(b) Light transfer in portrait photography scenes

Figure 9: **Visual Comparison between LumiNet and TransLight under different scenes.** Lu-miNet excels at transferring illumination status, whereas TransLight is better suited for transferring light effects themselves.

inference. We evaluate all 30 scenes in the test set by computing the RMSE and SSIM between the generated images and the reference images, and report the average scores. The results are shown in the Table 5. We also provide visualization comparison on MIIW (Murmann et al., 2019) dataset in Figure. 8. Although our method is not specifically designed for indoor scenes, it nevertheless demonstrates strong generalization capability.

Moreover, we also provide a qualitative comparison with LumiNet in Figure. 9. We conduct comparisons on two distinct tasks: state transfer of lamps in indoor scenes and light effect transfer in

UJpe: W3, W5

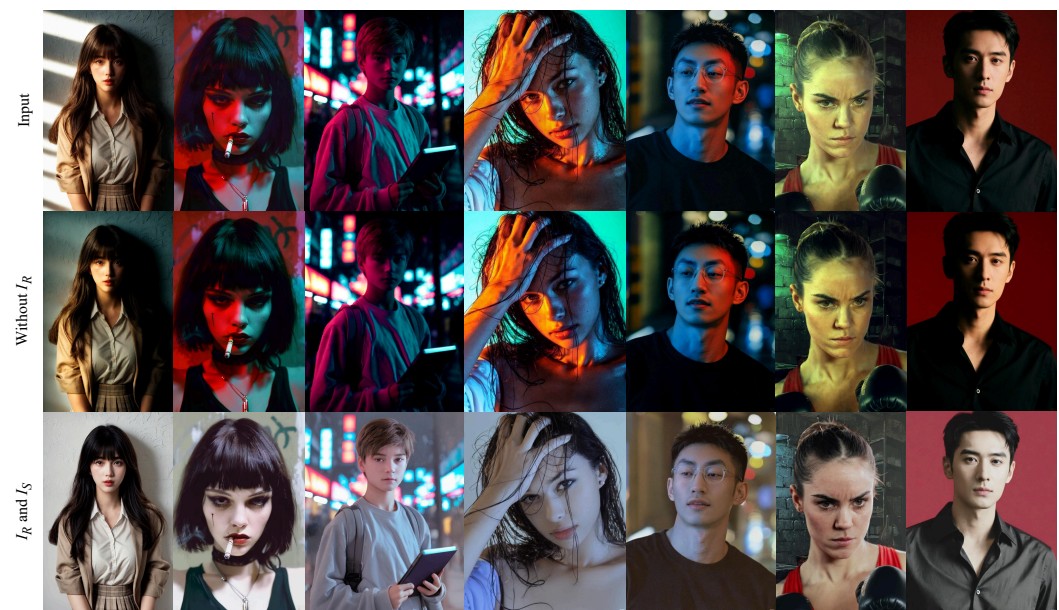

Figure 10: **Visual Comparison of Light Removal Results.** The first row shows the input images, the second row presents the results generated by the model trained without using $I_R$, and the third row displays the results generated by the model trained with both $I_S$ and $I_R$.

portrait photography scenes. As evidenced by the presented results, it is evident that our TransLight and LumiNet are fundamentally designed for different tasks. Our TransLight can not change the lamp in the input image from the off state to the on state based on a reference image. Similarly, LumiNet cannot add an iridescent halo surrounding the person to a portrait photo either.

UJpe: W4, Q4

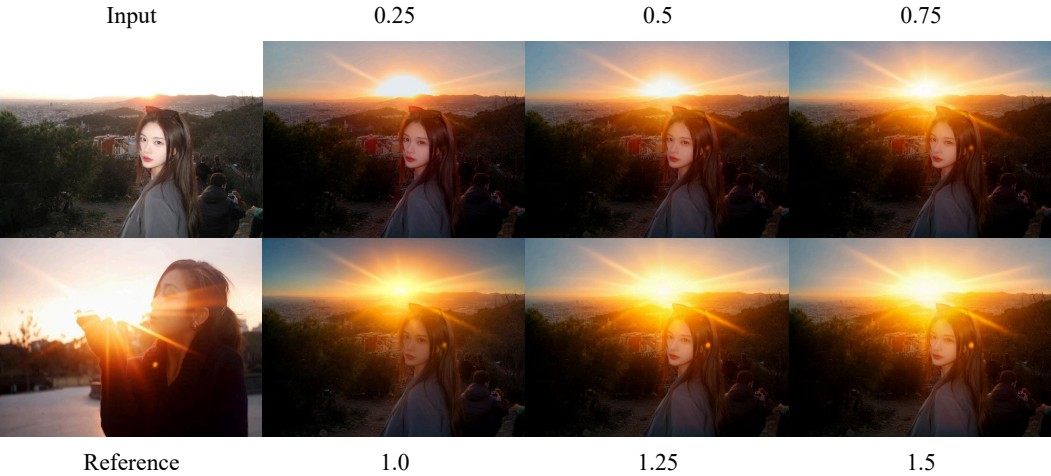

Figure 11: **Transferred results under different intensity coefficients**

## C  MORE VISUALIZATIONS

During the training of the light removal model, in addition to the synthesized image $I_S$, the relighted image $I_R$ is also provided as a conditional input. We employ relighting methods to simulate strong illumination contrasts present in real-world scenes, such as shadows and highlights, with the aim of enhancing the generalization capability of the light removal model in eliminating complex light effects. To intuitively illustrate the role of $I_R$, we present in Figure .10 the generated results of the

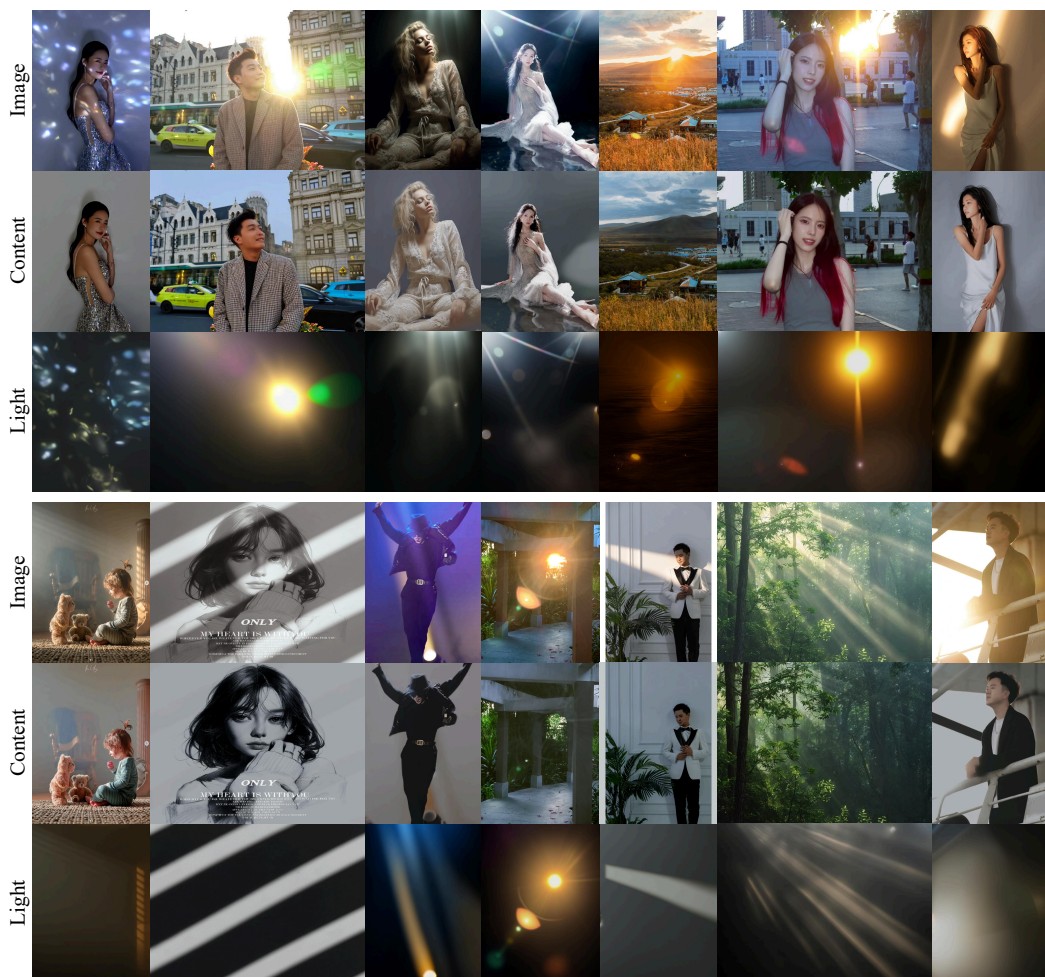

Figure 12: **Visualizations of our generative decoupling.** Our light removal model can eliminate light effects from images while preserving the underlying content unchanged as shown in the second line. Additionally, our light extraction model is capable of isolating the light effects from the image without introducing other extraneous objects.

light removal model trained with and without the inclusion of $I_R$. When $I_R$ is included as training data, the light removal model is able to simultaneously remove light effects as well as shadows and highlights on the subject's face.

We present more visualizations of our generative decoupling in Figure 12. In Figure 13, we show more qualitative comparison results of our TransLight on the task of light effect transfer. Our TransLight accurately captures the structural details of light effects and effectively transfers them onto the input image. Moreover, TransLight demonstrates superior performance in preserving content consistency and significantly outperforms existing methods such as IC-Light and Seedream 4.0 in mitigating content leakage. In addition to controlling the position and direction of light effects, our TransLight further enables adjustment of their intensity in the transferred results. We show an example in Figure 11.

UJpe: Q6;
SUh4: W4

# D  LIMITATION AND FAILURE CASE

The primary limitation of our TransLight framework lies in its dependence on the performance of the light extraction model. The quality of the final transferred output is heavily influenced by how effectively the light effect is extracted from the reference image. This implies that when the

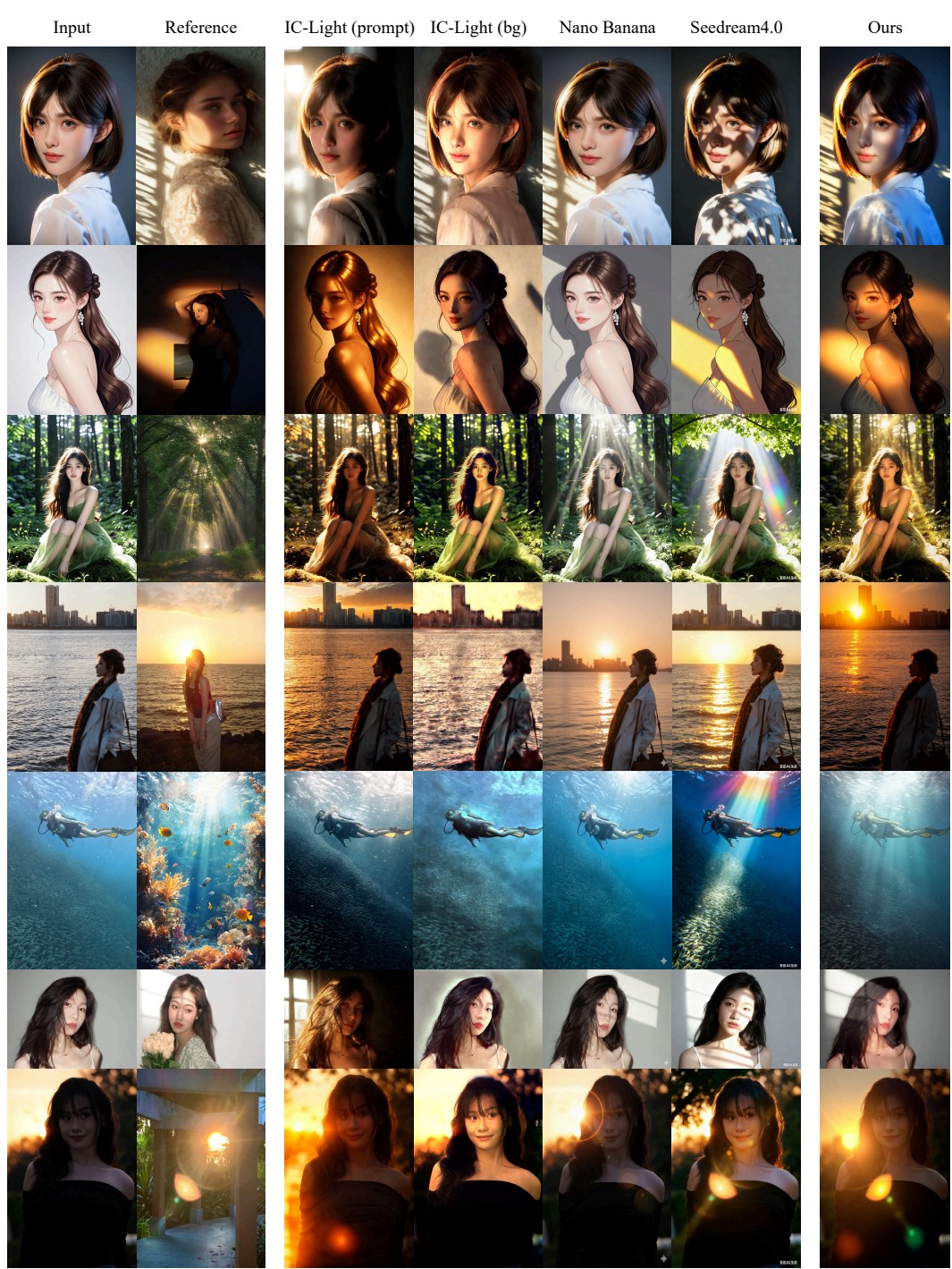

Figure 13: **Visualization comparison.** IC-Light (prompt) uses its prompt-conditioned model, and the prompt is derived from the reference image. IC-Light (bg) uses its background-conditioned model. For Nano Banana and Seedream 4.0, we provide both the input image and the reference image to the image editing model.

light effect in the user-provided reference is subtle or indistinct, our method may fail to produce satisfactory results. We show some failure cases in Figure. 14. When the light effect in the reference image is subtle or when the light extraction model fails to adequately capture the lighting effect, this may result in less pronounced transfer outcomes. In addition, when the reference image itself

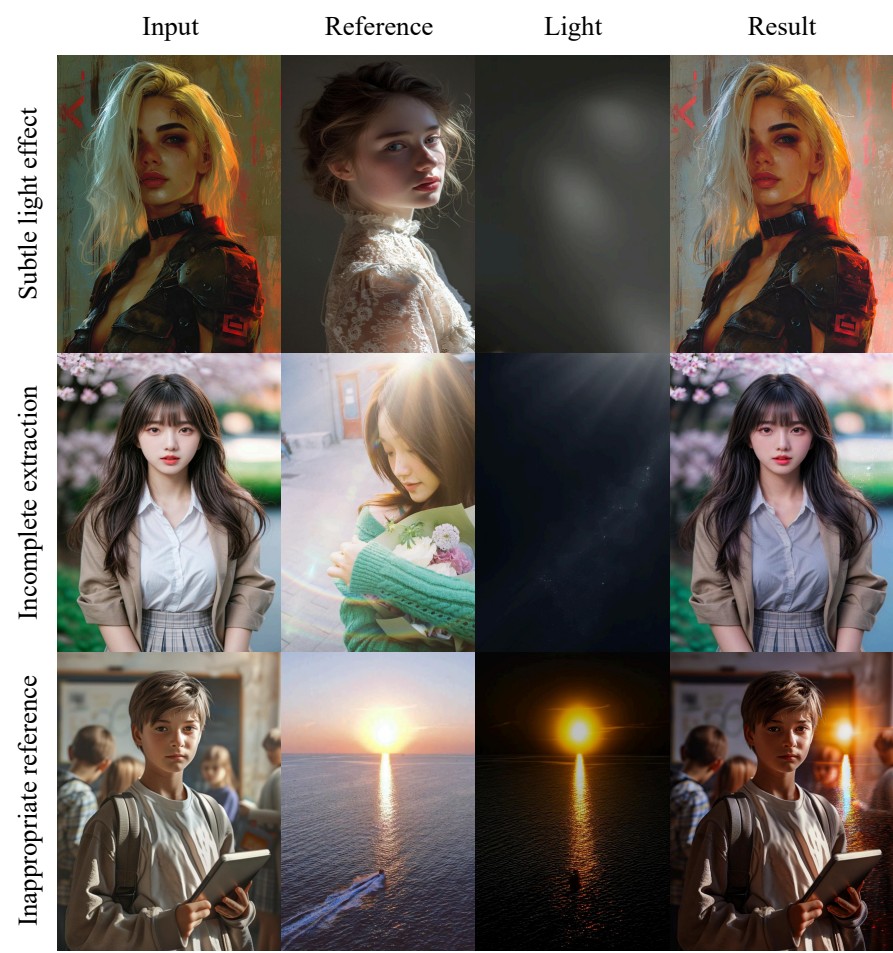

Figure 14: **Failure cases.** For subtle light effect case, the transfer degree is not obvious. For incomplete extraction case, since it is impossible to ensure 100% perfect disentanglement of light effects, the generated results may fall short of expectations. For inappropriate reference case, large scene mismatch may be meaningless for the light effect transfer task we defined, and it is unnecessary to pay too much attention to such case.

deviates too much from the input image scene, forced transfer may also lead to inappropriate results. This is because the light effect transfer task in such cases is inherently unreasonable. Given that the primary limitation of our method lies in the light extraction model's capacity to accurately capture light effects, a promising direction for future work in the context of light effect transfer is to further enhance the model's ability to disentangle and represent light effects.

67f2: Q4;
UJpe: W6, Q5;
SUh4: W3

## E  DISCUSSION

In this paper, we define a novel task that has not yet received significant attention: light effect transfer. This task aims to transfer the light effects from a reference image onto a target image. We think such a capability has substantial potential in real-world image-editing scenarios. For instance, after capturing a portrait, users may wish to enhance the visual appeal through appropriate illumination editing. Notably, users typically desire to preserve the original image content while incorporating artistic light effects, which may be sourced from high-quality image templates found on the internet or social media platforms. Motivated by this practical need, we propose TransLight.

Our approach presents a feasible solution to the light effect transfer task by first decoupling the light effect from the reference image into a standalone light-only image, and then compositing it onto the

target image. This strategy significantly mitigates the risk of content leakage, and by adjusting the position and orientation of the extracted light image, enables highly flexible and controllable light transfer with great ease. Although the transfer performance of our TransLight is less pronounced when the lighting effect in the reference image is subtle, this does not diminish the method's practical applicability or potential for real-world deployment. We have extensively validated our method across diverse reference image scenarios, demonstrating that it achieves lighting editing outcomes that are beyond the reach of existing illumination or image editing techniques. We believe that light effect transfer possesses a broad spectrum of application scenarios and represents a task eminently worthy of in-depth exploration and research. Furthermore, we are confident that our TransLight framework offers a plausible and valuable direction for advancing this field of study.

### E.1 Statement on The Usage of Large Language Models (LLMs)

In this paper, we employ Qwen 3 (Yang et al., 2025) and Gemini 2.5 flash (Google DeepMind, 2025) to perform grammatical revision and linguistic refinement of the manuscript. We ensure that the accuracy and validity of the theoretical explanations and conclusions presented in this paper are not influenced by LLMs.

