# OpenReview forum: "TransLight: Image-Guided Customized Lighting Control with Generative Decoupling"
_ICLR.cc/2026/Conference — ICLR 2026 Conference Withdrawn Submission_

### Official Review · Reviewer_SUh4 · 2025-10-29

**Soundness:** 3
**Presentation:** 3
**Contribution:** 3
**Rating:** 6
**Confidence:** 4

**Summary:**

This paper presents TransLight, a novel framework for image-guided light effect transfer, which enables high-fidelity transfer of complex lighting effects from a reference image to a target image while preserving content consistency. The core lies in the proposed Generative Decoupling strategy, which employs two diffusion models to disentangle content and light effects in real images, thereby constructing a large-scale dataset of image–content–light triplets. Having these data, the authors design a two-stage training framework: the first stage applies LoRA fine-tuning to preserve content integrity, while the second stage leverages ControlNet to learn light effect injection and geometric control. The final model supports controllable transformations of lighting effects, including translation, rotation, and intensity adjustment, achieving customizable illumination transfer that surpasses existing methods in both Light-FID and perceptual quality.

**Strengths:**

- The paper introduces a new problem, image-guided light effect transfer, which is distinct from traditional relighting or style transfer tasks, demonstrating originality and novelty.
- The proposed Generative Decoupling framework effectively separates content and light effects using two diffusion models, and the subsequent two-stage training with LoRA and ControlNet achieves a good balance between content stability and controllable lighting manipulation.
- The authors construct a large-scale dataset of over one million image-content-light triplets and conduct extensive experiments showing significant improvements over existing methods in both quantitative metrics and qualitative visual fidelity.
- The model supports flexible control of lighting position, direction, and intensity, and the results exhibit natural integration between content and light, indicating that the network has successfully learned a disentangled representation of lighting effects.

**Weaknesses:**

- The method focuses on perceptual-level lighting transfer (e.g., lens flares, streaks) without modeling scene geometry, material properties, or physically consistent illumination. As a result, the generated lighting effects may not correspond to physically plausible relighting outcomes.
- The million-scale triplet dataset is constructed through automatic generation and filtering using a VLM, yet the paper does not provide sufficient details about data diversity, quality control, or potential biases, and the cost of VLM is not cheap, raising concerns about reproducibility and dataset reliability.
- The light extraction module plays a key role in generating training triplets, but the paper does not analyze how imperfect or incomplete extraction affects TransLight’s training and final generation quality. No quantitative or failure-case analysis is provided.
- The paper presents relatively few qualitative examples compared to prior works such as IC-Light. The lack of diverse visual results weakens the empirical support for the method’s generality.

**Questions:**

In Section 3.3, Stage 1 is described as “a simple addition of the content image and the background light effects image.” Could the authors clarify whether the model receives this summed image as a single input, or two separate input channels (content + light concatenated)?

---

> ### Author Response · Authors · 2025-11-24
> **[1/2] Response to Reviewer SUh4**
>
> First of all, thank you very much for your opinions and suggestions. We will explain them one by one. You can also view the revised version, where we have marked the questions you raised to help you quickly locate them.
>
> ---
>
> **W1: The method focuses on perceptual-level lighting transfer (e.g., lens flares, streaks) without modeling scene geometry, material properties, or physically consistent illumination. As a result, the generated lighting effects may not correspond to physically plausible relighting outcomes.**
>
> We fully understand your concerns. Our method explicitly separates the light effects and then combines them onto other images. Although modeling scene geometry, material properties, or physically consistent illumination are not involved in this process, this does not mean that the images generated by our method will deviate from the real physical scenes. Firstly, the generative network IC-Light we use is trained on a wide range of data based on the principle of consistent light propagation, and it inherently has a strong ability to generate illumination. Secondly, our TransLight is trained with over a million triples of data, and the supervisory information is the illuminated images in real scenes. Appropriate training can enable our TransLight to fit the illumination distribution of these real images. Meanwhile, by expanding the training data, the distribution distance of the lighting effect between the generated results and the real images can also be further reduced.
>
> ---
>
> **W2: The million-scale triplet dataset is constructed through automatic generation and filtering using a VLM, yet the paper does not provide sufficient details about data diversity, ....**
>
> Thank you very much for your suggestion.
> - For cost: The triplet data construction process took approximately 12 days, and during this period, we used 16 L40s GPUs for all steps. It takes approximately one day to select images with strong light effects from massive data using VLM. It took 5 days and 6 days respectively to extract the light effects from these images and remove the light effects. Although InternVL2.5-8B acts as a referee and filter throughout the process, which is very crucial, the time consumption it generates is not the main part. Moreover, InternVL is completely open source and there are no additional fees.
>
> - For data diversity: We categorize all triplets in our dataset into nine light-effect types: [1.Light Beam in Tyndall Effect; 2.Dappled Light; 3.Backlighting Effect; 4.Iridescent Halo; 5.Lens Flare; 6.Bright Light Source; 7.Gleaming Water Reflection; 8.Projected Light Patch; 9.Volumetric Light Rays] . Using InternVL2.5-8B, we count the number of images associated with each category. In the revised version, we present the distribution as a histogram and provide representative examples for each category.
>
> - For quality control: We conduct experiments on the parameter $\gamma$ in the filtering step to provide more details. As shown in the table, we set $\gamma=0.98$ for the trade off between data quality and data quantity. The larger the threshold $\gamma$ is, the smaller the filtering force will be, the more training data there will be, and the poorer the overall data quality will be. Conversely, the lower the threshold $\gamma$, the greater the filtering force, the smaller the training data, and the higher the data quality. So we set $\gamma=0.98$ to conduct the experiment.
>
> | $\gamma$  |   Training samples    |   Light-FID   |
> |:---------:|:----------:|:---------:|
> |   1       |   ~1.5M               |   6.41        |
> |  0.99     |   ~1.3M              |   6.22        |
> |  0.98     |   ~1.2M               |   6.02        |
> |  0.97     |   ~0.9M               |   6.54        |
> |  0.96     |   ~0.6M               |   6.83        |
> |  0.95     |   ~0.5M               |   7.00        |
>
> - For potential biases: Our triplet dataset primarily encompasses two broad scene types: portrait photography and natural landscapes, with portrait scenes accounting for approximately 80\% of the data.
>
> More details about the triplet data types and distribution will be provided in the revised version.
>
> ---
>
> **W3: The light extraction module plays a key role in generating training triplets, but the paper does not analyze how imperfect or incomplete extraction affects TransLight’s training and final generation quality. No quantitative or failure-case analysis is provided.**
>
> We greatly appreciate your suggestion. As you said, the light extraction model does play a very crucial role. When it fails to accurately extract the light effect of the reference image, it will cause the transfer result to fail to meet expectations. This is indeed one of the main factors currently restricting the generation ability of our TransLight. We will provide specific fail cases in the revised version of the paper and conduct a detailed analysis.

---

> ### Author Response · Authors · 2025-11-24
> **[2/2] Response to Reviewer SUh4**
>
> **W4: The paper presents relatively few qualitative examples compared to prior works such as IC-Light. The lack of diverse visual results weakens the empirical support for the method’s generality.**
>
> We think your suggestion is very necessary. We will provide more visual comparisons in the revised version to demonstrate the superiority of our TransLight. We present the comparison of five methods: prompt-conditioned IC-Light, background-conditioned IC-Light, Nano Banana, Seedream 4.0, and our TransLight.
>
> ---
>
> **Q1: In Section 3.3, Stage 1 is described as “a simple addition of the content image and the background light effects image.” Could the authors clarify whether the model receives this summed image as a single input, or two separate input channels (content + light concatenated)?**
>
> In the stage 1, the input image received by Unet is a single image that has undergone a simple addition operation, which means it is a single input. The input channel is 8 of the first layer, we don't use additional channels. In this way, no structural modifications to the Unet of IC-Light are required.
>
> ---
>
> We sincerely thank you once again for your valuable suggestions. We hope our responses have addressed your concerns and clarified any confusion regarding the paper. Should you have any further questions, please do not hesitate to let us know.

---

### Official Review · Reviewer_UJpe · 2025-10-29

**Soundness:** 3
**Presentation:** 2
**Contribution:** 3
**Rating:** 2
**Confidence:** 4

**Summary:**

This paper transfers light effects from a reference to a target via “generative decoupling” with two diffusion models: a light-removal model producing content-only and a light-extraction model producing an effects-only map. They mine images with a VLM to form image/content/effects triplets and filter by features. Training is two-stage: (1) LoRA to enforce content preservation (suppress lighting), (2) ControlNet to inject the extracted effects map. At inference, the effects map is injected and can be rigidly transformed (translate/scale/flip) to harmonize illumination, not requiring explicit geometry or light-source modeling.

**Strengths:**

1) The method is simple and is an interesting decomposition into content vs. light effects

2) Scalable data pipeline for building triplets without manual labels.

3) Practical for creative workflows like image harmonization, where exact physical relighting is unnecessary.

4) The lighting removal is perhaps the most interesting part of the pipeline that can suppress strong lighting phenomena like glares, specular effects, etc.

**Weaknesses:**

1) One of the main claimed contributions of the paper is that it is“First to transfer light effects with geometric structure across disparate images,” which is not correct. Prior work already transfers lighting in a physically grounded way: Latent Intrinsics (cited) and its generative adaptation -- LumiNet: Latent Intrinsics meets diffusion models for indoor scene relighting (CVPR 2025; not cited) transfer lighting codes from a source to a target while handling large geometric mismatches in complex indoor scenes. Also, these comparisons are missing in the current paper.

2) While the generative decoupling construction is interesting, the method does not transfer lighting. It is illumination effects harmonization via a 2D effects layer. There is no discussion in the paper on how the proposed method differs from diffusion-based relighting methods that are physically motivated (like Latent Intrinsics, LumiNet, or Diffusion Renderer (CVPR 2025), which is again not cited in the paper.

3) No evaluation on standard relighting datasets (MIT Multi-Illumination, BIG-Time Time Lapse Dataset, etc.).

4) Does the method understand lighting semantics, such as transferring the lighting from switching on a light bulb in one scene to another? The proposed approach seems to be spatially biased, as it focuses on pixel-aligned matching and then harmonization between the source and target images, rather than employing global illumination reasoning.

5) “Light FID” is unvalidated for lighting structure; FIDs are known to be biased and are not a good measure of lighting phenomena. Missing structure-aware metrics using a standard dataset and a user study.

6) Missing limitations and failure examples.

**Questions:**

1) What exactly do authors mean by “light effects with geometric structure”? Which phenomena are included (cast shadows, speculars, bloom, flares, caustics), and which are excluded?
2) Can the authors elaborate on how this decomposition differs from traditional intrinsic image decomposition?
3) How does this approach differ technically from diffusion-based, physically motivated relighting methods (Latent Intrinsics, LumiNet, Diffusion Renderer)?

4) Can this method demonstrate source-level semantics (toggle a lamp, move a point light, change intensity/color) and report accuracy under controlled edits?

5) Under a large geometry mismatch (indoor vs outdoor), what are the failure modes of the 2D effects transfer? Please quantify and show examples. Also, how does the method perform on indoor scenes?

6) The paper mentions content leakage multiple times. If the extraction is “content-free,” is it possible to quantify the content leakage?

---

> ### Author Response · Authors · 2025-11-24
> **[1/3] Response to Reviewer UJpe**
>
> First of all, thank you very much for your opinions and suggestions. We will explain them one by one. You can also view the revised version, where we have marked the questions you raised to help you quickly locate them.
>
> ---
>
> **W1: One of the main claimed contributions...which is not correct...**
>
> We admit that the lack of comparison between LumiNet[1] and our method may indeed lead to questions about the originality of our method in the task of light effect transfer. However, LumiNet and our method have very obvious differences in application scenarios and generation effects. Our TransLight focuses on the artistic editing of light in portrait photography scenes, while LumiNet focuses on adjusting the status of indoor illumination. Our TransLight can fully refer to the specific geometric structure, direction and color of the light effect in the reference image, and replicate these features onto other images. LumiNet adjusts the illumination status of the input image by referring to the illumination status in the reference image. For example, when the lamp in the reference image emits warm-toned light, the lamp that is off in the input image will also become on. According to the visualization results provided in the LumiNet paper, it does not require that the geometric features of the light effect of the generated image be consistent with those of the reference image when performing transfer generation. We will cite and discuss LumiNet in the revised version of the paper.
>
> ---
>
> **W2: While the generative decoupling construction is interesting, the method does not transfer lighting...**
>
> First of all, we do not agree with what you said: "the method does not transfer lighting". When we discuss the light effect transfer, we do not wish to be confined to the overall transition of illumination status. We agree that converting the cold-toned light source in the indoor scene to a warm-toned light source consistent with the reference image is a kind of light effect transfer. Similarly, we also firmly believe that replicating the artistic light beams in the reference image onto other images is also a kind of light effect transfer.
>
> In terms of the generation target, our TransLight has a key difference from these methods. The target of these physically-based methods is to render the input image to a new illumination status, such as converting an "off" desk lamp to an "on" status based on the status of the reference image (Latent Intrinsics[2] and LumiNet[1]), or modifying the direction of the shadow from left to right according to a specific Env.Map (Diffusion Renderer[3]). Our TransLight focuses on adding customized light effects with distinct geometric structures to the input image. Moreover, our TransLight can also adjust the illumination status of the image by extracting light effects from the reference image to make the transfer result more realistic. For example, it can modify the shadow of the person's face according to the direction and position of the light effect, as shown in the example in the second sample of Figure 1. We will add detailed discussions about these methods in the revised version.
>
> ---
>
> **W3: No evaluation on standard relighting datasets (MIT Multi-Illumination, BIG-Time Time Lapse Dataset, etc.).**
>
> We agree that benchmarking on established public datasets is valuable for fair comparison.
> We conduct zero-shot evaluation experiments on the MIIW[4] dataset, as our model has not been trained or fine-tuned on this data. We follow the experimental setup of Latent-Intrinsic[2]. We randomly select an image and its 12 reference image for light effect transfer inference. We evaluate all 30 scenes in the test set by computing the RMSE and SSIM between the generated images and the reference images, and report the average scores. The results are shown in the following table.
>
> |   Methods                 |   Labels  |   RMSE↓   |   SSIM↑   |
> |:-------------:|:-----------:|:--------:|:---------:|
> |   Input Img               |   -       |   0.384   |   0.438   |
> |   SA-AE[5]                |   Light   |   0.288   |   0.484   |
> |   SA-AE[5]                |   -       |   0.443   |   0.300   |
> |   S3Net[6]                |   Depth   |   0.512   |   0.331   |
> |   S3Net[6]                |   -       |   0.499   |   0.336   |
> |   Latent-Intrinsic[2](σ=0)|   -       |   0.326   |   0.232   |
> |   Latent-Intrinsic[2]     |   -       |   0.297   |   0.473   |
> |   RGB-X[7]                |   -       |   0.256   |   0.476   |
> |   LumiNet[1]              |   -       |   0.180   |   0.647   |
> |   Ours                    |   -       |   0.212   |   0.627   |
>
> We need to emphasize once again that our TransLight is not designed for the transition of illumination status in indoor scenes. The evaluation results of our method on the MIIW dataset demonstrate its generalization capability, even though these metrics do not directly reflect the effectiveness of light effect transfer in portrait photography scenarios.

---

> ### Author Response · Authors · 2025-11-24
> **[2/3] Response to Reviewer UJpe**
>
> **W4: Does the method understand lighting semantics, such as transferring the lighting from switching on a light bulb...**
>
> Our method is actually not designed for semantic lighting, and thus is not suitable for switching the on/off statuss of table lamps in indoor scenes. However, we do not consider this a weakness that limits the wide application of our method in the future. The ability to transfer localized light effects from arbitrary references has strong applications in artistic editing, AR/VR, and photography enhancement. We once again state that there are very obvious differences in the design motivation, application scenarios and prediction targets of our TransLight and physically-based methods (Latent Intrinsics, LumiNet, Diffusion Renderer). Neither our TransLight nor IC-Light can change the lamp in the input image from the off status to the on status based on a reference image. Similarly, LumiNet and DiffusionRerender cannot add a rainbow light effect surrounding the person to a portrait photo either. You can get a more detailed explanation in the supplementary materials of the revised version.
>
> ---
>
> **W5: “Light FID” is unvalidated for lighting structure...**
>
> In addition to Light-FID, we also present the experimental results of structured indicators such as SSIM in Table 1. Additionally, we sincerely appreciate your suggestion regarding the user study. We conduct a manual evaluation along two dimensions: light effect consistency and content consistency. The results are summarized in the table below. For each sample, we present the generation results of five methods: prompt-conditioned IC-Light[8], background-conditioned IC-Light, Nano Banana[9], Seedream 4.0[10], and our TransLight, displayed in random order.
>
> |   Method              |  Light effect consistency |   Content consistency |
> |:----------:|:----------:|:----------:|
> |   IC-Light (prompt)   |       4.0\%               |       12.0\%          |
> |   IC-Light (bg)       |       8.4\%               |       8.8\%          |
> |   Nano Banana         |       39.5\%              |       61.3\%          |
> |   Seedream4.0         |       32.4\%              |       31.1\%          |
> |   Ours                |       84.0\%              |       73.8\%          |
>
>
> We will supplement the results on a MIIW dataset and the user study in the revised version.
>
> ---
>
> **W6: Missing limitations and failure examples.**
>
> Thanks for you suggestions. In fact, we have discussed the limitation of our method in the supplementary materials. In addition, we will provide failure cases and corresponding analyses in the revised version.
>
> ---
>
> **Q1: What exactly do authors mean by “light effects with geometric structure”?**
>
> “light effects with geometric structure” represents the visible light with a distinct form, including shape, structure, direction and color, etc. Common types include the Tyndall effect, projected light from windows, rainbow halos, and point light sources, among others. Among the categories you mentioned, bloom and flares are relatively common in our triplet dataset. We have analyzed the distribution of light effect categories across our million-scale triplet dataset and provide representative examples for each category. These results and visual samples can be found in the supplementary materials of the revised version, which we hope address your concerns regarding the type of light effects.
>
> ---
>
> **Q2: Can the authors elaborate on how this decomposition differs from traditional intrinsic image decomposition?**
>
> Physically-based methods, such as LumiNet, perform decomposition in the latent space, whereas our method operates directly in the pixel space. Our generative decoupling explicitly separates the light effects, fully preserving their geometric structure. It is highly suitable for image editing tasks that have strict requirements for the specific form of the light effects.
>
> ---
>
> **Q3: How does this approach differ technically from diffusion-based, physically motivated relighting methods (Latent Intrinsics, LumiNet, Diffusion Renderer)?**
>
> Technically, the primary distinction between our method and prior approaches is that we explicitly disentangle lighting effects in the pixel space. Our method does not require additional environment maps or HDR maps during image generation. Furthermore, by explicitly decoupling light effects into an independent RGB image, we enable flexible adjustments such as flipping and translation. The generative decoupling framework using two diffusion models is an innovative approach to address content-light entanglement, which has not been effectively resolved in prior work.
>
> ---
>
> **Q4: Can this method demonstrate source-level semantics (toggle a lamp, move a point light, change intensity/color) and report accuracy under controlled edits?**
>
> See above explanations in Weakness #4

---

> ### Author Response · Authors · 2025-11-24
> **[3/3] Response to Reviewer UJpe**
>
> **Q5: Under a large geometry mismatch (indoor vs outdoor), what are the failure modes of the 2D effects transfer? Please quantify and show examples. Also, how does the method perform on indoor scenes?**
>
> Our method is not designed for indoor scenes. Instead, it is tailored for portrait photography light editing tasks that have broad application prospects. When choosing reference images in portrait photography scenes, there is often a tendency towards scenes with the same type of template. Therefore, the concern about large geometry mismatch is unnecessary. For instance, it is inherently unreasonable to transfer the sunset glow from the seaside to an indoor photography scene. We will include similar examples in the failure case section of the revised version. For indoor scenes, we conduct evaluation experiments on the MIIW dataset. Additionally, in Figure 7 of the revised version, we provide a visualization of TransLight’s ability to switch a bedside lamp on and off in a bedroom scene.
>
> ---
>
> **Q6: The paper mentions content leakage multiple times. If the extraction is “content-free,” is it possible to quantify the content leakage?**
>
> We fully agree with your insightful suggestion. The effectiveness of our method in mitigating content leakage stems from its explicit disentanglement of light effects from the reference image. As shown in the user study table, the content consistency scores provide empirical evidence that our approach effectively preserves content integrity and prevents unintended content transfer. Moreover, the additional visual comparisons provided in the revised version further demonstrate that our method outperforms existing approaches in mitigating content leakage.
>
> ---
>
> [1] Xing X, Groh K, Karaoglu S, et al. Luminet: Latent intrinsics meets diffusion models for indoor scene relighting[C]//Proceedings of the Computer Vision and Pattern Recognition Conference. 2025: 442-452.
>
> [2] Xiao Zhang, William Gao, Seemandhar Jain, Michael Maire, David Forsyth, and Anand Bhattad. Latent intrinsics emerge from training to relight. In NeurIPS, 2024.
>
> [3] Liang R, Gojcic Z, Ling H, et al. Diffusion Renderer: Neural Inverse and Forward Rendering with Video Diffusion Models[C]//Proceedings of the Computer Vision and Pattern Recognition Conference. 2025: 26069-26080.
>
> [4] Lukas Murmann, Michael Gharbi, Miika Aittala, and Fredo Durand. A multi-illumination dataset of indoor object appearance. In ICCV, 2019.
>
> [5] Zhongyun Hu, Xin Huang, Yaning Li, and Qing Wang. Sa-ae for any-to-any relighting. In ECCV, pages 535–549. Springer, 2020.
>
> [6] Hao-Hsiang Yang, Wei-Ting Chen, and Sy-Yen Kuo. S3net: A single stream structure for depth guided image relighting. In CVPR, pages 276–283, 2021.
>
> [7] Zheng Zeng, Valentin Deschaintre, Iliyan Georgiev, Yannick Hold-Geoffroy, Yiwei Hu, Fujun Luan, Ling-Qi Yan, and Milos Ha ˇ san. Rgb¡-¿x: Image decomposition and synthesis using material-and lighting-aware diffusion models. In SIGGRAPH, pages 1–11, 2024.
>
> [8] Zhang L, Rao A, Agrawala M. Scaling in-the-wild training for diffusion-based illumination harmonization and editing by imposing consistent light transport[C]//The Thirteenth International Conference on Learning Representations. 2025.
>
> [9] Google DeepMind. Gemini 2.5. https://blog.google/technology/google-deepmind/gemini-model-thinking-updates-march-2025/, 2025.
>
> [10] Team Seedream, Yunpeng Chen, Yu Gao, Lixue Gong, Meng Guo, Qiushan Guo, Zhiyao Guo, Xiaoxia Hou, Weilin Huang, Yixuan Huang, et al. Seedream 4.0: Toward next-generation multimodal image generation. arXiv preprint arXiv:2509.20427, 2025.
>
> ---
>
> We sincerely thank you once again for your valuable suggestions. We hope our responses have addressed your concerns and clarified any confusion regarding the paper. Should you have any further questions, please do not hesitate to let us know.

---

> > ### Comment · Reviewer_UJpe · 2025-11-28
> > **Response to author's rebuttal**
> >
> > Thanks for the detailed response.
> >
> > 1) Can the authors share qualitative results on the MIIW dataset? Thanks for running the quantitative analysis.
> >
> > 2) Thanks to the authors for clarifying the scope of the paper -- it would be beneficial for the community if the authors can explicitly mention this scope "artistic editing of light in portrait photography scenes" and tone down the "Experimental results establish TransLight as the first method to successfully transfer light effects with geometric structure across disparate images, delivering more customized illumination control than existing techniques and charting new directions for research in illumination harmonization and editing."

---

> > > ### Author Response · Authors · 2025-11-29
> > > **Response to Reviewer UJpe**
> > >
> > > Thank you for your reply and suggestions.
> > >
> > >  1. In Figure 8 of the revised version, we present the visualization results of our method on the MIIW dataset and also compare them with Latent Intrinsic and LumiNet. Our method can accurately perceive the position and direction of the light effect in the reference figure, obtaining competitive results.
> > >
> > > 2. Thank you very much for your suggestion. We do believe that it is very important to clearly define the scope of application of our TransLight. Therefore, in the abstract, introduction and conclusion sections of the revised version, we have clearly stated that our TransLight focuses on portrait photography scenarios.
> > >
> > > We hope the revised version can address your concerns and confusion. If you have any further questions, please feel free to give us feedback at any time. Thank you again for your suggestions on our paper.

---

### Official Review · Reviewer_Y35F · 2025-11-01

**Soundness:** 3
**Presentation:** 3
**Contribution:** 3
**Rating:** 6
**Confidence:** 4

**Summary:**

This paper proposes a method to transfer light with two considerations: 1. the scene illumination and 2 the lighting effects. The two perspectives are explicitly decomposed with independent training data preparation.

**Strengths:**

1.	The problem modeling is good. Lighting effects are important for real photo but less discussed in relighting research.
2.	The data processing looks satisfying, and I encourage authors to release it.
3.	The model design looks reasonable and foreseeably easy to reproduce (if with proper data)

**Weaknesses:**

1.	From the results, it seems that rotational/directional illumination is less discussed. But I can understand that the main purpose is to achieve those lighting effects.
2.	The model seems relatively weak in changing the illumination direction in its input images – many results display the inability in changing light direction.
3.	Most results show very strong artistic lighting expressions but in real photos they are relatively rare, so the scope may be a bit narrow. But I understand that this is the whole point of this framework: to realize those lighting expressions.

**Questions:**

1. Why we need both  stage 1 and 2 in figure 2 c? Are they just ablation or do we have special considerations?

---

> ### Author Response · Authors · 2025-11-24
> **Response to Reviewer Y35F**
>
> First of all, thank you very much for your opinions and suggestions. We will explain them one by one. You can also view the revised version, where we have marked the questions you raised to help you quickly locate them.
>
> ---
>
> **W1: From the results, it seems that rotational/directional illumination is less discussed. But I can understand that the main purpose is to achieve those lighting effects.**
>
> The visualization results we present focus on demonstrating that our method has the ability to replicate artistic light effects with distinct geometric features. Our method is not specifically designed for the task of modifying the illumination status of indoor scenes (such as rotating the angle of shadows). We will provide a detailed discussion of the distinctions between our TransLight and existing illumination transfer methods for indoor scenes in the supplementary materials of the revised version.
>
> ---
>
> **W2: The model seems relatively weak in changing the illumination direction in its input images – many results display the inability in changing light direction.**
>
> Modifying the illumination direction effect in the input image by referring to the light effect in the reference figure is indeed challenging. Our method can achieve this effect to a certain extent. For instance, in the second example in Figure 1, the contrast of light and shade on the face of the person can be clearly seen, which very clearly reflects the direction of the lighting, and this is also where our method is superior to Nano Banana.
>
> ---
>
> **W3: Most results show very strong artistic lighting expressions but in real photos they are relatively rare, so the scope may be a bit narrow. But I understand that this is the whole point of this framework: to realize those lighting expressions.**
>
> What we pursue is customized and stunning light effect editing. Therefore, the light effects of the reference images we study are often structurally distinct and have a prominent artistic style. Although these light effects are not considered scenes in ordinary life settings, they are very useful for enhancing the artistic beauty of images, such as portrait photography and the construction of scene atmosphere. The ability to transfer localized light effects from arbitrary references has strong applications in artistic editing, AR/VR, and photography enhancement.
>
> ---
>
> **Q1: Why we need both stage 1 and 2 in figure 2 c? Are they just ablation or do we have special considerations?**
>
>  Two-stage training is necessary, and each stage is designed with its own motivation and reason. As described in Section 3.3, the lora fine-tuning in the first stage is to enable the model to keep the background unchanged. The original iclight model would remove the background of the input image and regenerate it during inference. However, this mode is not applicable to our task. The second stage of training is aimed at providing complete light effect images and enhancing the degree of conditional injection. Applying only one of the stages will affect the generation effect. Meanwhile, since the optimization goals of the two stages are different and lora fine-tuning requires fewer steps, we train the two stages separately. Here we present a comparison of the quantitative results of different training strategies.
>
> | Tranining strategy            |   Light-FID   |
> |:--------------:|:-----------------:|
> |   Only LoRA                   |    7.94       |
> |   Only ControlNet             |    8.44       |
> |   LoRA+ControlNet(joint)      |    7.42       |
> |   LoRA+ControlNet(seperate)   |    6.02       |
>
> ---
>
> We sincerely thank you once again for your valuable suggestions. We hope our responses have addressed your concerns and clarified any confusion regarding the paper. Should you have any further questions, please do not hesitate to let us know.

---

### Official Review · Reviewer_67f2 · 2025-11-01

**Soundness:** 3
**Presentation:** 3
**Contribution:** 3
**Rating:** 6
**Confidence:** 2

**Summary:**

This paper introduces TransLight, a novel framework for image-guided light effect transfer. The key goal is to transfer customized light effects with complex geometric structure from a reference image to a target image, while preserving the target image content. The core idea is a generative decoupling strategy, where two fine-tuned diffusion models are used to separate content and light in natural images. Using this, the authors construct over one million image-content-light triplets, which are then used to train a two-stage generation pipeline consisting of LoRA-based fine-tuning and ControlNet conditioning. The method outperforms existing relighting and editing techniques both quantitatively (e.g., Light-FID) and qualitatively.

**Strengths:**

- The formulation of the “light effect transfer” task—distinct from conventional relighting or style transfer—is novel and well-motivated. The generative decoupling framework using two diffusion models is an innovative approach to address content-light entanglement, which has not been effectively resolved in prior work.
- The pipeline is well-engineered, with thoughtful design choices such as DINOv2-based filtering and multi-source relighting in training. Ablation studies confirm the effectiveness of LoRA pretraining and data filtering. The method achieves impressive improvements over IC-Light in both PSNR/SSIM and Light-FID.
- The paper is generally well-written and logically organized. Figures are clear and informative, especially those visualizing decoupling and transfer effects. The proposed framework is well explained across training and inference phases.
- The ability to transfer localized light effects from arbitrary references has strong applications in artistic editing, AR/VR, and photography enhancement. This work may serve as a foundation for future light-aware generative modeling.

**Weaknesses:**

- Inconsistent training supervision for decoupling: The light removal model is trained with both synthesized (IS) and relighted images (IR), while the light extraction model uses only IS. This inconsistency is not justified and may lead to a domain gap in model robustness. A clearer rationale or ablation would strengthen the method section.
- Threshold γ lacks theoretical justification: The cosine similarity threshold γ=0.98 is crucial in triplet filtering. However, the choice seems empirical and lacks sensitivity analysis or theoretical lower bounds. It remains unclear how this affects recall vs. precision in data quality.
- Two-stage training not rigorously compared: While the authors adopt a two-stage pipeline (LoRA → ControlNet), there is no analysis of whether joint training or alternative one-stage schemes could achieve similar or better performance. The added complexity needs clearer justification.

**Questions:**

- Can the authors provide a quantitative comparison between the light extraction model trained with and without the IR augmentation used in the light removal model? How significant is the performance drop in complex lighting scenarios?
- How sensitive is the performance to the choice of threshold γ in the triplet filtering stage? Could the authors include a small ablation or discussion of the impact of lower or higher γ? Does the hard threshold γ have a lower limit?
- In inference, does TransLight use both LoRA and ControlNet models? Could a single unified network achieve similar results? How is the runtime efficiency impacted?
- How does TransLight perform when the reference image contains subtle or ambient lighting effects rather than high-intensity structured light?
- Would using a unified prompt-to-light map generation strategy (e.g., text-guided diffusion) eliminate the need for large-scale triplet construction?

---

> ### Author Response · Authors · 2025-11-24
> **[1/2] Response to Reviewer 67f2**
>
> First of all, thank you very much for your opinions and suggestions. We will explain them one by one. You can also view the revised version, where we have marked the questions you raised to help you quickly locate them.
>
> ---
>
> **W1: Inconsistent training supervision for decoupling**
>
> When training the light extraction model, we need an appropriate target image as the supervised signal, and this target image is the light effect material image $L$ that we have prepared in advance as shown in Figure 2 (a). In fact, $I_R$ is not suitable for training the light extraction model because we cannot obtain the corresponding light effect image $L$. To enhance the robustness and the generalization capability of the light extraction model, when synthesizing $I_S$, we perform various data augmentation operations on L, such as flipping, cropping, and random masking, etc.
>
> ---
>
> **W2: Threshold γ lacks theoretical justification**
>
> In our filtering step, the significance of setting the threshold $\gamma$ is to ensure that there is an obvious difference between the no light image $I$ and the natural image $I_L$. This helps the model effectively learn the influence of the light effect on the image, thereby achieving efficient convergence. We set $\gamma=0.98$ for the trade off between data quality and data quantity. The larger the threshold $\gamma$ is, the more data remains after filtering, but the data quality varies. Conversely, the smaller the threshold $\gamma$ is, the less data remains after filtering, but the data quality is higher. We present the data quantity and experimental results corresponding to different thresholds $\gamma$ in the following table:
>
> | $\gamma$  |   Training samples    |   Light-FID   |
> |:---------:|:---------------------:|:-------------:|
> |   1       |   ~1.5M               |   6.41        |
> |  0.99     |   ~1.3M              |   6.22        |
> |  0.98     |   ~1.2M               |   6.02        |
> |  0.97     |   ~0.9M               |   6.54        |
> |  0.96     |   ~0.6M               |   6.83        |
> |  0.95     |   ~0.5M               |   7.00        |
>
> ---
>
> **W3: Two-stage training not rigorously compared**
>
> Two-stage training is essential, and each stage is designed with its own motivation and reason.
> - As described in Section 3.3, the lora fine-tuning in the first stage is to enable the model to keep the background unchanged. The original iclight model would remove the background of the input image and regenerate it during inference. However, this mode is not applicable to our task.
> - The second stage of training is aimed at providing complete light effect images and enhancing the degree of conditional injection.
>
> Applying only one of the stages will affect the generation effect. Meanwhile, since the optimization goals of the two stages are different and lora fine-tuning requires fewer steps, we train the two stages separately. The experimental results also prove that joint training is difficult to fully optimize the model. Here we present a comparison of the quantitative results of different training strategies.
>
> | Tranining strategy            |   Light-FID   |
> |:---------:|:-------------:|
> |   Only LoRA                   |    7.94       |
> |   Only ControlNet             |    8.44       |
> |   LoRA+ControlNet(joint)      |    7.42       |
> |   Lora+ControlNet(seperate)   |    6.02       |
>
> ---
>
> **Q1: Can the authors provide a quantitative comparison...**
>
> As statusd above, due to lack of the supervisory signal, $I_R$ cannot be used in the training of the light extraction model directly.
>
> ---
>
> **Q2: How sensitive is the performance to the choice of threshold γ...**
>
> We have provided a detailed explanation in "W2". If the threshold $\gamma$ is too low, it will result in too fewer remaining data after filtering, which may affect the training effectiveness of the model.
>
> ---
>
> **Q3: In inference, does TransLight use both LoRA and ControlNet models?**
>
> Yes, we use both LoRA and ControlNet during inference stage. Using only a single module will limit the transfer effect, that is, the transferred light effect will not be obvious enough. The time consumption brought by the lora module and the controlnet module is 0.6s . We present the results in the following table, and the resolution of the test sample is 768×1024. The latency in the last row of the table includes the latency of extracting the light effect image from the reference image about 1.1s.
>
> |   Method                  |   Latency |
> |:---------:|:-------------:|
> |   IC-Light(baseline)      |   3.7222s |
> |   +LoRA                   |   3.8803s |
> |   +LoRA+ControlNet        |   4.3924s |
> |   TransLight              |   5.5067s |

---

> ### Author Response · Authors · 2025-11-24
> **[2/2] Response to Reviewer 67f2**
>
> **Q4: How does TransLight perform when the reference image contains subtle or ambient lighting effects rather than high-intensity structured light?**
>
> When the light effects in the reference image have no obvious structure or no obvious light source, Translight may fail to capture the obvious light effects, resulting in insignificant changes in the transfer results. However, this is not an ideal application scenario for light effect transfer tasks.
>
> One potential solution of adding subtle or ambient light effects is to use textual description which is not what our paper focuses on.
> For further intuitive explanation, we will provide some failure cases when the light effect of the reference image is subtle in the revised version.
>
> ---
>
> **Q5: Would using a unified prompt-to-light map generation strategy (e.g., text-guided diffusion) eliminate the need for large-scale triplet construction?**
>
>  We believe that triplet data is highly necessary for specific tasks. Its significance lies in providing strong supervision for the model, especially the geometric structure features of light effects, which are difficult to obtain merely through text description.
>
> ---
>
> We sincerely thank you once again for your valuable suggestions. We hope our responses have addressed your concerns and clarified any confusion regarding the paper. Should you have any further questions, please do not hesitate to let us know.

---

### Author Response · Authors · 2025-11-27
**Rebuttal summary and changes in the revised version**

We are extremely grateful to the reviewers for the constructive opinions and suggestions. In the revised version, we have supplemented more experimental results and provided detailed explanations to address the concerns and doubts of the reviewers. Meanwhile, for the convenience of verification, we have marked the newly added content in blue font and attached the reviewers names (67f2, Y35F, UJpe, SUh4) beside it.

----

The key updates are as follows：

- **Comparison with physically-based indoor scenes relighting methods:** We add citations and discussion of indoor scene methods in **Section 2.3**. In **Appendix B**, we compare our method with these approaches in detail, conduct quantitative evaluations on the MIIW dataset, and provide visual comparisons with LumiNet across different scenes.

- **Triplet data distribution:** We categorize our million-scale triplet dataset by light effect and report the statistical results and visualization examples in **Appendix A.2**.

- **More ablation study:** We add the ablation of filtering threshold $\gamma$ and training strategy of two stages in **Section 4.5**.

- **User study:** We present the user study results in **Section 4.4**.

- **More visualization comparison:** We add more visual comparison with IC-Light, Nano Banana and Seedream 4.0 in **Appendix C**.

- **Limitation and failure case:** In **Appendix D**, we add an analysis of the limitations of our method, and also provide visualization results of the failure cases.

---

We once again thank the reviewers for their insightful comments. We hope that our explanations and revised version will address all remaining concerns.

---

### Note · Authors · 2026-01-27

**Comment:**

It is very regrettable that even though we provided sufficient additional explanations, our paper still was not accepted. We have decided to withdrawal the paper.

**Withdrawal Confirmation:**

I have read and agree with the venue's withdrawal policy on behalf of myself and my co-authors.

---

### Meta-Review · Area_Chair_eQw2 · 2026-01-14

**Summary:**

This paper introduces TransLight, a framework that enables transfer of light effects. Reviewers' major concerns focus on: 1. The lack of comparison with physically-based indoor scenes relighting methods. 2. Reason of using two-stage training. 3. Ability to transfer of ambient lighting or subtle lighting. 4. Inability to change lighting direction. 5. The limited scope that focuses on very strong artistic lighting expressions.

The authors addresses a portion of these concerns, and the remaining are not convincingly tackled. Therefore, I suggest to reject this paper.

**Reviewer Concerns:**

The reasons to use two-stage training vs. one-stage training have been addressed in the rebuttal. However, from my perspective, some important concerns such as the inability to change lighting direction and the limited scope are not fully addressed.

**Reviewer Scores:**

Some reviewers participated in the discussion after the rebuttal, but did not change the score.

---

### Decision · Program_Chairs · 2026-01-26

Reject